# Cardiac cAMP-PKA Signaling Compartmentalization in Myocardial Infarction

**DOI:** 10.3390/cells10040922

**Published:** 2021-04-16

**Authors:** Anne-Sophie Colombe, Guillaume Pidoux

**Affiliations:** INSERM, UMR-S 1180, Signalisation et Physiopathologie Cardiovasculaire, Université Paris-Saclay, 92296 Châtenay-Malabry, France; anne-sophie.colombe@inserm.fr

**Keywords:** heart, myocardial infarction, cardiomyocytes, cAMP signaling, A-kinase anchoring protein, protein kinase A, phosphodiesterases

## Abstract

Under physiological conditions, cAMP signaling plays a key role in the regulation of cardiac function. Activation of this intracellular signaling pathway mirrors cardiomyocyte adaptation to various extracellular stimuli. Extracellular ligand binding to seven-transmembrane receptors (also known as GPCRs) with G proteins and adenylyl cyclases (ACs) modulate the intracellular cAMP content. Subsequently, this second messenger triggers activation of specific intracellular downstream effectors that ensure a proper cellular response. Therefore, it is essential for the cell to keep the cAMP signaling highly regulated in space and time. The temporal regulation depends on the activity of ACs and phosphodiesterases. By scaffolding key components of the cAMP signaling machinery, A-kinase anchoring proteins (AKAPs) coordinate both the spatial and temporal regulation. Myocardial infarction is one of the major causes of death in industrialized countries and is characterized by a prolonged cardiac ischemia. This leads to irreversible cardiomyocyte death and impairs cardiac function. Regardless of its causes, a chronic activation of cardiac cAMP signaling is established to compensate this loss. While this adaptation is primarily beneficial for contractile function, it turns out, in the long run, to be deleterious. This review compiles current knowledge about cardiac cAMP compartmentalization under physiological conditions and post-myocardial infarction when it appears to be profoundly impaired.

## 1. Introduction

The beating heart ensures an essential function by supplying an oxygen- and nutrient-rich blood flow to the organs. Efficient blood flow distribution in both the pulmonary and systemic circulations allows the exchange of oxygen and carbon dioxide. Pump function is maintained by the automatic cardiac excitation that spontaneously triggers rhythmic and periodic contractions. This dynamic process is modulated in response to various physiological and pathophysiological stimuli such as hormones, neurotransmitters, cardiac injuries and stresses that adapt/maladapt heart function [1,2]. Cardiac injuries and stresses are defined as events leading to cardiac diseases (e.g., medication, pollution, xenohormones, hypertension and myocardial infarction (MI)). Under physiological conditions, the sympathetic nervous system releases catecholamines (e.g., epinephrine and norepinephrine), which bind to α- or β-adreno receptors (α- or β-AR) and adapt the cardiac function by increasing the heart rate (HR), stroke volume and cardiac output. In contrast, acetylcholine produced by the parasympathetic system exerts opposite effects to those of catecholamines [1]. Like many other hormones and neurotransmitters affecting cardiac function, catecholamines and acetylcholine bind to and activate G protein-coupled receptors (GPCRs), which modulate intracellular cAMP (cyclic adenosine 3′,5′-monophosphate) signaling [3]. The intracellular concentration of this second messenger is finely tuned between the activities of adenylyl cyclases (ACs) and phosphodiesterases (PDEs), which synthetize and catalyze cAMP, respectively [4]. Cyclic AMP mediates the cellular response adaption to extracellular stimuli after activation of several specific downstream effectors such as protein kinase A (PKA), cAMP-dependent exchange protein (Epac), cyclic nucleotide-gated channels (CNGC) and Popeye domain-containing (POPDC) proteins [5]. Therefore, to ensure a high level of specificity in the cellular response to distinct stimuli, cAMP signaling needs to be tightly regulated in space and time. This is resolved by compartmentalized cAMP nanodomains that encompass signaling effectors (e.g., kinases, phosphatases and channels) and modulators (i.e., ACs and PDEs) [5]. The organization of these intracellular cAMP nanodomains is orchestrated by A-kinase anchoring proteins (AKAPs) [4].

Evidence of cardiac cAMP signaling compartmentalization relies on experiments performed in the late 1970s [6,7,8]. A heart perfused with isoproterenol (a β-AR agonist) raises the strength of contraction (i.e., inotropic response) and increases the myocardial cAMP level. The latter mobilizes PKA, which, in turn, mediates cascade activation of the phosphorylase kinase and the glycogen phosphorylase, triggering glycogen breakdown. Although PGE1 (agonist of prostaglandin E1 receptor) perfusion in the heart increases cAMP production and PKA activity, it failed to regulate cardiac contractile function and glycogen metabolism [7,9]. At the cellular level, adult rat ventricular myocytes stimulated with glucagon, glucagon like peptide-1 and β2-AR agonist exhibit, for all, an increase in the intracellular cAMP content but distinct cellular responses that are specific to each stimulus. Activation of β2-AR enhances the positive inotropic response, while the glucagon exerts positive inotropic and lusitropic (i.e., rate of myocardial relaxation) effects. In contrast, glucagon like peptide-1 ensures a modest negative inotropic response [10,11,12,13]. Therefore, restraining the cardiac cAMP signaling to distinct intracellular compartments provides means to achieve specific cardiomyocytes’ adaptation to various extracellular stimuli. Therefore, the concept of cardiac cAMP signaling compartmentalization was established (for review, see [14]).

Ischemic heart diseases or coronary heart diseases (CHDs) are the leading causes of death in industrialized countries and kill 15 million people worldwide each year [15]. CHD is a syndrome that defines heart complications with total or partial coronary artery occlusion leading to heart muscle ischemia, and blood and oxygen deprivation. CHD contains several heart conditions with various clinical features such as acute coronary syndrome (ACS), chronic angina pectoris (CAP) and chronic ischemic heart disease (CIHD) [16]. ACS, which includes unstable angina, acute myocardial infarction (AMI; with or without ST segment elevation (i.e., STEMI and non-STEMI)) and sudden cardiac death (SCD), needs intensive and acute treatment, while patients with CAP (i.e., stable angina, Prinzmetal’s variant angina, Mazzeri’s angina and angina with normal coronary arteries) or CIHD (i.e., “silent” ischemia, heart failure and ischemic cardiomyopathy) undergo chronic treatments. A prolonged cardiac ischemia characterizes MI usually triggered after a blood clot, coronary artery constriction or atherosclerosis and leads to irreversible cardiomyocyte death, affecting cardiac function [17]. To compensate the impaired cardiac function observed post-MI, the body activates an adaptive mechanism that includes the chronic sympathetic nervous system activation [18]. While this adaptation is primarily beneficial for contractile function, the chronic sympathetic activation in the long run turns out to be deleterious [18]. These harmful effects are mediated by the chronic activation of the β-AR/cAMP signaling cascade, which causes adverse cardiac remodeling, cardiac myocyte death and fibrosis, leading to fatal heart failure and arrhythmias [18,19]. Currently, there is no therapeutic option to reverse CHD or MI and mid-term mortality remains high [15]. Nevertheless, immediate treatments (e.g., angioplasty, surgery) and medication (e.g., β-blockers, angiotensin converting enzyme (ACE)) improve patient’s prognosis and quality of life [20,21,22]. Among medications, β-blockers are a preeminent therapy limiting the severity of heart damages through β-AR competitive antagonization and inhibition of chronic cAMP signaling activation [13,19,20,23,24]. However, β-blockers cause severe side effects and are ineffective in a large proportion of patients [25,26,27,28]. In this context, several studies have attempted to develop new alternative therapeutic strategies to β-blockers. Therefore, some emerging ideas propose selectively targeting cAMP modulators or effectors as innovative curatives for cardiac diseases (e.g., MI, heart failure). These would offer similar benefits to β-blockers without their side effects [29,30,31,32]. As cAMP signaling is a key regulator of cardiac physiology, this review aims at describing molecular events that mediate cAMP signaling compartmentalization in cardiomyocytes under physiological and pathophysiological conditions of MI.

## 2. G Protein-Coupled Receptors (GPCRs)

This section details the current knowledge about major cardiac GPCRs that directly modulate cAMP signaling and for which the regulation of cardiac function under physiological conditions and post-MI has been reported (Table 1). Other cardiac GPCRs and their function have been reviewed in [3].

### 2.1. Structure

GPCRs are seven-transmembrane domain receptors with an extracellular N-terminal extremity and an intracellular C-terminal tail, which activates heterotrimeric G proteins consisting of G_α_ and G_β,γ_. In the inactive state, these three subunits form a heterotrimer with GDP-bound G_α_. Upon GPCR stimulation, GTP replaces GDP, which leads to a conformational change and releases the Gα subunit. The nature of the G_α_ protein (Gα_s_, Gα_i/o_, Gα_q_) and G_β,γ_ ensure specificity in the activation of intracellular signaling. GTP hydrolyzation in GDP by the G_α_ subunit triggers GPCR inactivation [52].

### 2.2. Adrenoreceptors

#### 2.2.1. α-Adrenergic Receptors (α-AR)

Among α-ARs, α1-AR and α2-AR are involved in the regulation of cardiac function. The α1-AR signals through Gα_q_ stimulation and phospholipase C activation [53]. In presynaptic nerves, α2-AR coupled to Gα_i/o_ inhibits AC activity and reduces catecholamine release (Table 1) [54]. There is also evidence of α2-AR expression in cardiomyocytes that, under stimulation, mediates negative chronotropic (i.e., HR) and inotropic responses [33]. In a dog model of MI with left anterior descending (LAD) coronary artery ligation, the inhibition of α2-AR increases susceptibility to arrhythmia. Although the underlying mechanism of these antiarrhythmic properties remains unclear, it could involve α2-AR located in Purkinje fibers [34].

#### 2.2.2. β-Adrenergic Receptors (β-AR)

In the heart, three β-AR isotypes are co-expressed (i.e., β1-AR, β2-AR and β3-AR), activated by catecholamines and coupled to small Gα_s_ or Gα_i_ proteins. While Gα_s_ activation triggers cAMP signaling, Gα_i_ inhibits AC activity and reduces the intracellular cAMP content. In cardiomyocytes, the β1-AR isoform is predominant and represents about 80% of total β-AR, and the β2-AR isoform represents approximately 20%, whereas β3-AR is poorly expressed [35,55]. β1-AR and β2-AR are mainly coupled to Gα_s_ and promote an increase in cAMP production. In addition, β2-AR can couple to Gα_i_ to reduce cAMP production and thus to temporize chronotropic effects mediated by prolonged Gα_s_ activation [6,35]. Although β1-AR and β2-AR activations induce common cardiac effects, they also display specific responses due to their distinct subcellular localization. β1-AR is ubiquitously distributed across the plasmalemma, while cardiac β2-AR expression is restricted to the T-tubules [35]. Under physiologic conditions, β1-AR adapts cardiac contractile function through positive regulation of inotropic, chronotropic and lusitropic effects (Table 1). Similarly, β2-AR mediates positive inotropic responses. However, β2-AR regulation of the cardiac lusitropic effect remains controversial (Table 1). In highly structured adult rat ventricular cardiomyocytes, β2-AR restricted to T-tubules failed to trigger a positive lusitropic response, contrary to immature neonatal rat ventricular myocytes [35]. Interestingly, transgenic mice with β1-AR and/or β2-AR KO point to β1-AR as being sufficient and necessary to maintain catecholamine-induced positive inotropic and chronotropic effects in vivo [36]. The classic model of β-AR activation depicts initiation of the signaling from the plasmalemma; however, this has been challenged by recent findings reporting also pools of active β1-AR/Gα_s_ and β2-AR/Gα_s_ inside cells at the Golgi apparatus and endosomes [56,57]. The endosome-localized β2-AR ensures specific transcriptional activity, while functions associated with β1-AR distributed at the Golgi apparatus remain unexplored [58]. These findings add an additional level of complexity to the cAMP signaling compartmentalization and should be further investigated under cardiac pathophysiological conditions, as they may offer new potential therapeutic targets.

Corresponding to their specific subcellular localizations, β-ARs exhibit distinct effects on cardiac function post-MI. β1-AR induces cardiac dysfunction and cardiomyocyte apoptosis post-MI [36]. Furthermore, specific β1-AR inhibition with siRNA decreases cardiomyocyte apoptosis in the ischemic area and improves cardiac function in a rat model of MI with LAD ligation [59]. In contrast, β2-AR exhibits cardioprotective properties. In acute MI, stimulation of β2-AR during reperfusion limits infarct size and the release of cardiomyocyte injury markers such as circulating troponin I (TnI) [38]. Moreover, it also reduces deleterious cardiac remodeling and protects cardiac function in a PKA-Akt-eNOS pathway activation-dependent fashion [38]. In an ischemia/reperfusion (I/R) injury model, the stimulation of β2-AR before reperfusion partly inhibits the inflammatory response. This triggers the release of interleukin-10 (IL10), reducing leukocyte activation, which is known to increase infarct size during reperfusion [39]. However, another study reported that post-chronic MI, neither β1 nor β2-AR influences infarction size [36]. This discrepancy could be explained by differences in the MI model studied (i.e., I/R vs. permanent ligation) and/or the timing of analysis post-surgery (i.e., acute vs. chronic) [36,39]. In chronic MI, β2-AR redistributes to the plasma membrane, reflecting T-tubule network disorganization that appears in the heart failure (HF) state [60]. Although cardiac β2-AR density remains preserved, β1-AR expression decreases at the infarct and border areas [37].

GRK2 (G protein-coupled receptor kinase 2 or βARK1: β-adrenergic receptor kinase1) mediates β-AR desensitization and internalization. While the β-AR desensitization process protects the heart after acute MI, it also contributes to maintaining the vicious circle of β-AR stimulation that leads to HF [61]. Therefore, the inhibition of the desensitization process with intravenous delivery of adenoviral constructs that directs the expression of a GRK2 antagonist improves left ventricular cardiac function in a rat model of chronic MI [62]. These promising data pave the way for a putative new therapy to treat chronic MI patients.

Among β-AR gene polymorphisms reported, patients exhibiting the haplotype G16R, Q27E or T164I in *ADRB2* (i.e., gene coding for β2-AR) show a reduction in MI incidence. In contrast, several *ADRB1* gene variants (i.e., coding for β1-AR) in combination with β-blocker therapy display higher risks [63,64,65].

β3-AR is coupled to both Gα_s_ and Gα_i_ and in contrast to the other β-ARs, it ensures a negative cardiac inotropic effect [40,55]. Post-MI, the β3-AR expression level increases and shows cardioprotective properties [41]. In acute MI, pre-activation of β3-AR before the reperfusion limits the infarct size. Its activation in chronic MI protects cardiac function and reduces fibrosis [41,42]. These effects are partly mediated by the activation of β3-AR-Akt-eNOS signaling that delays mitochondrial mPTP opening and increases cell survival [42]. In addition, nNOS also appears to be involved in β3-AR stimulation-induced cardioprotection [41].

### 2.3. Miscellaneous Cardiac GPCRs

#### 2.3.1. Muscarinic Receptors Type 2

Muscarinic receptor type 2 (M_2_R) couples to Gα_i_ and binds acetylcholine to mediate vagal regulation of cardiac function. M_2_R activation inhibits adenylyl cyclase activity, which depresses cAMP production and triggers a negative chronotropic effect (Table 1) [35]. Although M_2_R is upregulated at the remote zone of human MI patients, its expression remains unchanged in the infarct area. It has been proposed that this upregulation could prevent arrhythmogenicity induced by sympathetic activity [43]. Similarly to β-AR, M_2_R exhibits polymorphisms that are associated with increased risks of cardiac death following MI [66].

#### 2.3.2. Adenosine Receptors (or P1 Receptors)

Four different genes encode for the adenosine receptor (AR) isoforms (i.e., A_1_AR, A_2A_AR, A_2B_AR, A_3_AR), which are all expressed in the heart. A_2A_AR and A_2B_AR isoforms couple to Gα_s_ and their stimulation produces intracellular cAMP. In contrast, A_1_AR and A_3_AR couple to Gα_i/o_ and show opposite effects to those of A_2_AR isoforms. Post-MI, ARs influence cardiac fibrosis and exhibit cardioprotective effects, which has been reviewed (Table 1) [44,45].

#### 2.3.3. Prostaglandin Receptors

In the heart, several prostaglandin receptor isotypes co-exist, which modulate the cAMP signaling differently. Prostaglandin EP2 and EP4 receptors coupled to Gα_s_ increase AC activity and cAMP production, while prostaglandin EP3 receptor, which is coupled to Gα_i_, depresses cAMP signaling [35]. Post-MI, the expression levels of prostaglandin EP3 and EP4 receptors increase and ensure cardioprotective properties (Table 1) [25]. Therefore, the stimulation of prostaglandin EP4 receptor by PGE_2_ in an I/R model induces cardioprotective effects through the increase in IL10 secretion and inhibition of TNFα [67]. Gene therapy overexpressing prostaglandin EP4 receptor in mice subjected to MI restores cardiac function and reduces hypertrophy and fibrosis [46]. In addition, prostaglandin EP3 receptor selective activation reduces the infarct size post-MI [47].

#### 2.3.4. Glucagon and Glucagon Like Petide-1 Receptors (GCCR and GLP1R)

Glucagon and Glucagon Like Peptide-1 (GLP1) are known to regulate glucose homeostasis and metabolism. In the heart, GCCR (glucagon receptor) couples Gα_s_ or Gα_i_ to modulate the cAMP production. Therefore, GCCR stimulation triggers a positive inotropic response that is blunted after Gα_i_ coupling [48]. In MI, GCCR stimulation exacerbates the cardiac disease by increasing the apoptosis, the infarct size and the mortality rate (Table 1) [49]. Therefore, specific GCCR inhibition exhibits beneficial effects on ventricular remodeling, cardiac metabolism and function.

Upon stimulation, GLP1R triggers cAMP production, which ensures a modest negative inotropic response (Table 1). This specific GLP1 compartmentalized signaling decreases myofilament affinity for calcium [13]. Interestingly, the injection of GLP1 or a GLP1R agonist (liraglutide) before MI activates pro-survival kinase and reduces the infarct size [50,51].

## 3. Adenylyl Cyclases

### 3.1. Structure and Function

Adenylyl cyclases (ACs) are a family of enzymes that mediate intracellular cAMP production from available ATP. In mammals, nine plasmalemma-bound isoforms (i.e., AC1–9) and one soluble isoform (i.e., sAC or AC10) have been reported [3]. Stimulated GPCRs directly regulate ACs throughout heterotrimeric G proteins (e.g., Gα_s_ or Gα_i/o_). GTP-bound alpha subunit of Gs (Gα_s_) stimulates all AC isoforms to produce cAMP, while Gα_i/o_ inhibits the activity of several ACs (i.e., AC1, 3, 5–6 and 8–9). Plasmalemma-bound ACs are structured with two hydrophobic TM1 and TM2 domains of six transmembrane segments each. TM1 and TM2 are linked together by an intracellular C1 loop which forms the catalytic site by dimerization with the C2-terminal sequence (for review, see [29,68,69]). Soluble AC lacks the transmembrane TM regions but contains the catalytic pocket (formed by C1 and C2 domains) and regulator clusters located on the C-terminal sequence. Heterotrimeric G proteins do not modulate sAC activity, which is reported to be stimulated by calcium and bicarbonate to produce cAMP [68,69]. sAC resides in the nucleus, centriole and mitochondria [70].

Although there is consensus in the scientific community about AC1–7, 9 and sAC expression in the heart, cardiac AC8 expression remains somewhat controversial [28,71]. It is noteworthy that AC5 and AC6 represent the most abundant cardiac isoforms.

### 3.2. Role of Major ACs in Cardiac Physiology and During MI

Studies on knockout (KO) mice models helped to decipher the roles of AC5 and AC6 in cardiac function [72,73,74]. Upon stimulation of left ventricular cardiomyocytes, AC5 and AC6 account for approximately 35% and 60% of intracellular cAMP production, respectively [73,75]. Although AC5 and AC6 exhibit some redundancy, their respective compartmentation in cardiomyocytes also provides some specificity in their regulation of cardiac function. AC5 locates at the T-tubule network and forms a complex with caveolins to mediate the β1- and β2-AR response [74]. In contrast, AC6 distributes with β1-AR on the outside of the T-tubule network [74]. It is known that cardiac AC5 promotes inotropic and chronotropic positive responses. Under β-AR stimulation, AC5 raises intracellular cAMP levels, activating downstream effectors, which, in turn, accelerates the HR, amplifies cardiomyocyte fractional shortening and increases left ventricle pressure [72,76]. However, the implication of AC5 and AC6 in the regulation of basal contractile function remains unclear [73,76]. Similarly to AC5, AC6 also modulates the inotropic response [73].

In cardiomyocytes, the specific subcellular compartmentation provides means to achieve regulation of AC5 and AC6 activity. PDEs which strongly regulate AC5 activity in T-tubules modulate, to a lesser extent, AC6 located at the plasmalemma [74]. In the basal condition, a constitutively active Gα_i_ inhibits AC6 to maintain low cAMP levels [77]. However, under β1-AR stimulation, this inhibition is relieved, which triggers cAMP production [77]. In contrast, AC5 turns out to be inhibited by Gα_i_ post-β-AR activation (see Section 2.2.2).

Interestingly, cardiac fibroblasts (CF) and cardiomyocytes display a diminution in the AC expression level as well as activity after MI [78,79]. Post-MI, CF cause fibrosis and scar formation at the infarct zone [80]. It has been reported that the diminution of the AC5 and AC6 expression level in CF exacerbates collagen deposition under β-AR stimulation in a chronic MI rat model. This suggests an AC-induced antifibrotic function that would prevent deleterious cardiac remodeling after MI [81]. It is commonly recognized that AC5 participates actively in the adverse cardiac remodeling, while AC6 offers cardioprotective effects post-MI [82]. Therefore, therapeutics targeting AC-mediated cAMP production are of interest to counteract either the chronic β-AR-induced cardiac remodeling or to limit infarction. Accordingly, infusion of the AC5 inhibitor vidarabine (i.e., AraA) or C90 before or shortly after coronary reperfusion reduces β-AR signaling and infarct size in an I/R mouse model [83,84]. However, the efficacy of AC5 inhibition by vidarabine remains controversial [83,85]. Similarly, in vitro inhibition of AC5 by PMC-6 prevents cardiomyocyte apoptosis without affecting the cell contractility under chronic β-AR stimulation [86]. Strategies aiming at increasing AC6 activity exhibit beneficial effects in MI outcomes. Mice overexpressing AC6 reveal a better survival rate than wild type (WT) mice post-MI. In addition to the recovery in cardiac contractility and relaxation, these transgenic AC6 mice also exhibit reductions in left ventricular dilatation and AV block incidence [87]. Moreover, in a model of MI-induced chronic heart failure, AC6 overexpression maintains the cardiac ejection fraction, prevents myocyte apoptosis and improves cardiac function [88].

### 3.3. Miscellaneous Cardiac ACs

In sinoatrial node (SAN), the cardiac pacemaker activity ensures a normal heart rhythm and rate. Therefore, SAN dysfunction leads to HR disorders. Arrhythmias are common features post-MI (e.g., bradycardias, tachycardias, heart blocks) and are symptomatic of poor patient prognosis, triggering cardiac arrests and sudden deaths [89]. In SAN, calcium induces AC1 activation, which, in turn, produces cAMP and modulates the pacemaker *I*_f_ current [90]. Therefore, the modulation of AC1 activity in SAN has been proposed to prevent arrhythmia occurrence. In a model of atrioventricular-blocked dogs, the overexpression of AC1 by gene therapy in SAN restored pacemaker activity and sensitivity to neurohormonal regulation [91].

In CF, AC3, 5 and 6 are expressed principally in caveolae. Interestingly, in these cells, angiotensin 2 (ANG II) potentiates the β-AR response through AC3 activity [92]. ANG II binds to AT1 receptor and activates Gα_q_, which increases the intracellular calcium level. This induces AC3-dependent cAMP production which comes in addition to that produced by β-AR stimulation. Together, the accumulated cAMP production limits collagen synthesis [92].

AC9 represents a minor source of total cardiomyocyte cAMP production (<3%) compared to the major activity of AC5/6. However, AC9 KO mice develop bradycardia and diastolic dysfunction with a preserved ejection fraction [82]. In cardiomyocytes, a pool of AC9 complexes with *AKAP9*, PKA and the potassium channel KCNQ1 to favor cardiomyocyte repolarization after sympathetic stimulation (see Section 5.3) [82,93,94]. In addition, the same group reported that cardiac AC9 regulates Hsp20 phosphorylation levels [82]. Interestingly, PKA-mediated Hsp20 phosphorylation shows cardioprotective properties after cardiac I/R injury (see Section 6.2) [95].

sAC produces cAMP in the mitochondrial matrix, which triggers PKA-dependent phosphorylation of cytochrome-c, regulating ATP production [96,97]. In myocardial infarction, reperfusion leads to cardiomyocyte injury and death associated with mitochondrial dysfunction. A study performed in models mimicking I/R in cardiomyocytes highlighted sAC as a key regulator of PKA-induced Bax phosphorylation, which triggers its mitochondrial translocation, leading to cell apoptosis [86]. Controversially, a recent study reported mitochondrial sAC overexpression to exhibit cardioprotective properties, improving cardiomyocyte survival after I/R [96].

## 4. Cyclic-AMP Downstream Effectors

### 4.1. Protein Kinase-A or cAMP-Dependent Protein Kinase (PKA)

#### 4.1.1. Structure

In the heart, Protein Kinase-A (PKA) is the major cAMP downstream effector. The PKA holoenzyme is a heterotetramer composed of two regulatory (R) and two catalytic (C) subunits. R subunits contain an N-terminal docking and dimerization (D/D) domain, a linker region with a PKA C subunit inhibitor site and two cAMP binding sites (i.e., CNB-A and CNB-B). The D/D domain locates within the first 45 residues of each R subunit, which mediates R dimerization and A-kinase anchoring protein (AKAP) docking [98]. The PKA R inhibitor site consists of phosphorylation motifs that inhibit the PKA C subunit in a substrate/autoinhibitory manner [99]. Cyclic AMP molecules bind first to CNB-B sites that promote an intramolecular steric change and facilitate cAMP binding to CNB-A sites, which trigger PKA C activation [100]. The PKA C subunit phosphorylates serine and threonine residues in consensus amino acid sequences of types: Arg-Arg-X-Ser/Thr; Arg-Lys-X-Ser/Thr; Lys-Arg-X-Ser/Thr; or Lys-Lys-X-Ser/Thr (with X as hydrophobic residues) [101]. The classic binary model of PKA activation depicts the release of C subunits from the holoenzyme to phosphorylate PKA-dependent residues. However, a recent study indicated that an active PKA C subunit remains leashed to the PKA holoenzyme [102]. This suggests the phosphorylation of PKA substrates to occur within a restricted nano-compartment of 15–25 nm range inside the cell [91]. Therefore, PKA subcellular compartmentalization appears essential to ensure selective phosphorylation of its substrates (see Section 4.). Two classes of PKA holoenzymes have been identified (i.e., type I and II), which differ in their R subunits (RI and RII) [103,104]. Furthermore, each PKA subunit (RI, RII and C) exists as multiple isoforms (RIα, RIβ, RIIα, RIIβ, Cα, Cβ, Cγ and PRKX) that are encoded by distinct genes. While type I PKA is classically known to be mainly cytosolic, type II associates with organelles and specific cellular structures [4]. In the heart, the RIα, RIIα and Cα isoforms are expressed, as well as, to a lesser extent, Cβ [105].

#### 4.1.2. PKA Modulates Cardiac Function

PKA plays a central role in the regulation of cardiac function and adaptation to stress (e.g., cardiac excitation–contraction coupling (CEC)). Upon β-AR stimulation and cAMP production, PKA phosphorylates CEC substrates and modulates positive chronotropic, inotropic and lusitropic responses [3]. In SAN, PKA triggers the positive chronotropic effect by regulating the calcium clock. This process is mediated by PKA-induced phosphorylation of L-type calcium channel (LTCC), ryanodine receptor (RyR) and phospholamban (PLB), which increase calcium release and HR [106]. Upon β-AR stimulation, these substrates turn out to also be phosphorylated by PKA in ventricular cardiomyocytes, increasing the contraction force and leading to a positive inotropic effect (Figure 1) [106]. Moreover, the positive lusitropic effect is under the control of PKA-dependent myofilament protein phosphorylation such as troponin I (TnI) and Myosin Binding Protein-C (MyBP-C) (Figure 1) [106]. Acute PKA activation mediates cardiac adaptation (e.g., fight-or-flight response). However, a chronic activation triggers adverse cardiac remodeling [18].

#### 4.1.3. PKA in Myocardial Infarction

PKA expression and activity post-MI remain unclear. Several studies reported an increase in type I PKA expression in acute MI in a rat model [107]. Studies performed in an I/R mice model described a decrease in type I PKA expression, while PKA C and RII subunits’ expression remained unchanged [108]. However, several studies reported neither modifications in PKA expression nor activity [109,110]. These discrepancies could be explained by models used and or by the time of study post-MI. Promptly after stress, β-AR signaling is mobilized, increasing PKA activity and pacing the heart pump counterbalancing MI-induced cardiac dysfunction [111,112]. However, sustained β-AR stimulation leads to GPCR desensitization, a compensatory mechanism diminishing PKA activity. Interestingly, PKA signaling has been described to exhibit cardioprotective effects post-MI. PKA activation in the pre-ischemic heart as a preconditioning condition limits infarct size though Rho-Kinase inhibition [113]. The antioxidant N-acetyl cysteine (NAC) shows antiarrhythmic properties. These are mediated after activation of glutathione that triggers ACs-induced cAMP production and PKA activity. Thereafter, PKA inhibits GSK3β, which abolishes connexin-43 (Cx43) internalization and occurrence of arrythmias [114]. Furthermore, adrenomedullin administration in Langendorff-perfused hearts limits infarct size in a cAMP/PKA signaling pathway activation-dependent manner favoring mitoK*_Ca_* channel opening [115]. However, the molecular mechanisms underlying these processes need to be further investigated. In additional studies, it has been reported that sitagliptin (i.e., dipeptidyl-peptidase-IV inhibitor), pioglitazon (i.e., PPAR-γ agonist) and simvastatin (a statins family member) limit infarct size in a PKA-dependent manner [112,116,117].

### 4.2. Epac (Exchange Protein Activated by cAMP)

#### 4.2.1. Epac Structure

Epac is a guanine nucleotide exchange factor (GEF) protein that activates Ras superfamily small GTPases Rap1 and Rap2. Two Epac isoforms (i.e., Epac1 and Epac2) are expressed in the heart. The Epac expression pattern changes during the life span. While Epac2 expression is predominant in adults, Epac1 is the major neonatal isoform [118]. The Epac N-terminus displays a regulatory region that contains one CNB (or two for Epac2) similar to that of the PKA regulatory subunit, which activates Epac upon cAMP binding. Furthermore, this region also exhibits a Disheveled/Egl-10/Pleckstrin (DEP) domain that favors Epac membrane localization. The catalytic C-terminal extremity contains three domains (i.e., REM, RA and CdC25-HD). The REM (Ras Exchange motif) domain stabilizes the Epac CdC25-HD catalytic helix and may scaffold additional regulatory proteins [119]. The Ras-association (RA) domain interacts with GTP-bound Ras and mediates Epac localization at the plasma membrane. The CdC25 homology GEF (CdC25-HD) domain triggers Epac guanine-nucleotide exchange activity [119].

#### 4.2.2. Epac Cardiac Function

Epac enhances cardiac contractility though regulation of calcium signaling [120]. Epac activation with the specific cAMP agonist 8-CPT-2′-O-Me-cAMP (i.e., 007) mobilizes the Rap1-PLC-CAMKII signaling pathway. In this context, the activation of CAMKII phosphorylates RyR2 and induces Ca^2+^ released from the sarcoplasmic reticulum (SR) that is independent of PKA and the LTCC-mediated calcium-induced calcium release process (CICR) (Figure 1) [120]. Epac can also activate PKC that phosphorylates cardiac TnI and MyBPC, which enhance cell shortening (Figure 1) [120]. However, the Epac-mediated positive inotropic effect remains controversial. These divergences rely on stimulation duration, as chronic Epac stimulation induces SR calcium depletion and reduction in calcium transients [120]. In the heart, it is established that specific Epac isoform compartmentalization mediates distinct subcellular effects. While Epac2 locates at the T-tubules network and regulates CAMKII and PKC signaling, peri-nuclear Epac1 modulates gene transcription (Figure 1) [121].

Finally, the role of Epac in cardiac pathophysiology remains ambiguous as it has been reported to have effects that are both pro- and anti-apoptotic, both pro- and anti-fibrotic and both pro- and anti-hypertrophic, with pro-arrhythmic effects [122,123]. These discrepancies need to be further elucidated.

#### 4.2.3. Epac in Myocardial Infarction

Post-MI, the profibrogenic factor TGF-β1 decreases Epac1 expression in cardiac fibroblasts located at the border zone, which inhibits fibroblast migration and induces collagen synthesis [124]. In contrast, Epac1 overexpression suppresses TGF-β1-induced collagen synthesis. Although the molecular mechanisms remain elusive and need to be further underpinned, Epac1 activation is reported to improve cardiac function by limiting infarct size, preventing atrial fibrosis, cardiac hypertrophy and remodeling [125]. Interestingly, a cell therapy approach in which mesenchymal stem cells (MSCs) are transplanted into the infarction zone to heal the diseased cardiac part needs activation of Epac-Rap1 signaling [126]. Epac activation favors homing, adhesion and differentiation of MSCs in cardiomyocytes which improves heart morphology, prevents ventricular dilatation and rescues cardiac contractility [126]. In redundancy with PKA (see above), NAC also mediates activation of Epac signaling that inhibits GSK3, limits Cx43 down-regulation and reduces arrythmia occurrence post-MI [114].

### 4.3. Cyclic Nucleotide-Regulated Cations Channels (CNCC)

CNCC are a heterogeneous superfamily of ion channels activated by cAMP or cGMP (Figure 1). The heart expresses two subtypes of this superfamily, which include hyperpolarization-activated cyclic nucleotide-gated ion (i.e., HCN) and cyclic nucleotide-gated ion (i.e., CNG) channels. Both HCN and CNG channels assemble at the plasmalemma in tetramer complexes. The transmembrane core consists of alpha-helical segments that form the ion-conducting pore, which in the case of the HCN channel exhibits a supplementary voltage sensor domain conferring specifically to this channel voltage-dependent gating properties [127]. The C-terminal extremity of HCN and CNG contains a CNB domain that binds cAMP and cGMP. Interestingly, HCN channels exhibit a higher affinity for cAMP binding than cGMP and vice versa for CNG channels. It is noteworthy that cyclic nucleotides binding is not required for HCN channel activation; however, it modifies channel voltage-dependent activation to more positive membrane potentials [127]. Both CNG and HCN channels are permeant to monovalent cations (i.e., Na^+^ and K^+^) and Ca^2+^ with distinct properties (Figure 1). The four types of HCN channels (HCN1–4) are expressed in the heart [128]. Among them, HCN4 is preferentially expressed in SAN, while HCN2 appears to be the major subtype in ventricles [129]. In SAN, HCN channels regulate the I*_f_* current and trigger cardiac pacemaker activity. Therefore, HCN4 KO mice develop severe bradycardia and arrhythmic events [128]. Dynamic changes in HCN expression occurring post-MI have been reported. While the HCN4 expression level increases, favoring I*_f_* current and arrhythmia occurrences in the left ventricular myocardium post-MI, the HCN1 expression level, however, diminishes [130]. The ivabradine therapeutic prescribed in patients to reduce arrhythmic events also inhibits the I*_f_* current and the increase in HCN4 expression post-MI [131]. Alternatively, spironolactone that targets HCN4 expression has also been proposed as a post-MI anti-arrhythmic medication [132].

### 4.4. Popeye Domain-Containing Protein

Popeye domain-containing (POPDC) proteins are plasmalemma proteins that are encoded by three different genes (Figure 1) [133]. POPDC proteins consist of a short extracellular N-glycosylated N-terminal domain linked to three transmembrane domains. The cytosolic tail harbors the POPDC and a variable C-terminal domain (CTD) that can be phosphorylated (e.g., under β-AR stimulation). Interestingly, POPDC functions as a high-affinity cAMP binding site that differs from consensus CNB domains found in PKA, Epac and HCN. POPDC proteins are expressed in the heart and particularly in the conduction system (i.e., SAN and atrio ventricular nodes) [133]. POPDC knockout mice exhibit a normal basal HR but show sinus bradycardia under stress response with structural alterations of SAN myocytes [134]. Deletion of POPDC leads to cardiac arrhythmia in zebrafish [135,136]. Interestingly, patients harboring POPDC mutations present cardiac arrhythmias [135,137,138]. Post-I/R injury, POPDC-null mice show a larger infarct size and bad cardiac recovery compared with wild-type mice [139].

## 5. A-Kinase Anchoring Proteins (AKAP)

### 5.1. Structure and Function

The spatial regulation of intracellular PKA signaling is orchestrated by its compartmentalization to a precise subcellular subset mediated by direct interaction with A-kinase anchoring proteins (AKAPs) [140]. AKAPs are a large family of structurally distinct proteins (>50 identified) that are defined by their ability to provide spatial and temporal regulation of PKA signaling events. To belong to the AKAP family, proteins must exhibit two specific properties: (i) must contain an A-kinase binding (AKB) domain that consists of a 14–18 amino acid amphipathic helix region, which interacts with the dimerized PKA R-subunit D/D domain; (ii) must contain a unique subcellular targeting domain that directs PKA/AKAP complexes to a defined location inside the cell (i.e., organelle, plasmalemma, protein complexes or lipids). In addition to these features, it has been reported that several AKAPs are directly bound to or in the close vicinity of PKA-specific substrates and that they could assemble signalosome complexes by direct interactions with additional multivalent signaling proteins (e.g., kinases, PDEs, phosphatases (PPs), Epac) [4,141,142,143]. Integration of PPs and PDEs into PKA/AKAP complexes provides a supplementary regulation level with a tight control of the cAMP–PKA signaling termination.

AKAPs, which are widely expressed in various cell types and tissues, form precise macromolecular complexes to ensure the specificity in PKA-dependent signaling cascades [4,144,145,146]. Historically, it was known that most AKAPs preferentially anchored the PKA type II holoenzyme with a high affinity, while few displayed dual-specificity with the capacity of binding both PKA R subtypes [4,147,148]. In the last decade, a new class of AKAPs has been discovered and reported to specifically bind PKA type I (e.g., SKIP, smAKAP) [4,142,149,150]. AKAPs were first named according to their respective molecular mass. Although widely approved, this nomenclature turned out to be confusing due to ortholog AKAP genes encoding for proteins with various molecular masses between species. Therefore, in an attempt to standardize the nomenclature, AKAPs are now classified according to the HUGO gene nomenclature committee name (Table 2).

### 5.2. Cardiac AKAPs with Patophysiological Function in MI

*AKAP1* (i.e., D-AKAP1 or S-AKAP84, AKAP121, AKAP149) is a dual-specificity AKAP that distributes to the mitochondrial outer membrane (Table 2). The first 30 amino acids of the protein sequence correspond to a mitochondrial targeting sequence. In addition to the AKB domain, *AKAP1* exhibits a PP1 (Protein Phosphatase 1) binding sequence and a KH-Tudor domain that restricts and favors translation of specific mRNAs (e.g., Star, *SOD2*, *F0-f*). Furthermore, *AKAP1* anchors CaN (calcineurin, PP2B), PDE4 and Drp1 (GTPase Dynamin-related protein 1) to regulate mitochondrial fission and fusion (Table 2; Figure 2A) [140]. *AKAP1* coordinates a cardioprotective macromolecular complex mediating PKA-dependent Drp1 phosphorylation, which, in turn, abolishes Drp1/FIS1 interaction and inhibits mitochondrial fission, leading to cell survival (Figure 2A) [173]. This process is counterbalanced by CaN recruitment to the *AKAP1* signaling complex, which, in contrast, favors Drp1 dephosphorylation and mitochondrial fragmentation. Furthermore, *AKAP1*-anchored PKA enhances BAD phosphorylation that abolishes BAD-Bcl2 interaction and prevents cardiomyocyte death (Table 2) [140]. *AKAP1* KO mice with permanent coronary ligation exhibit an extensive myocardial infarction size, illustrating the cardioprotective properties of this AKAP [174]. In MI, hypoxia activates Siah2 (ubiquitin-protein ligase seven in absentia homolog 2), which ubiquitinates *AKAP1* and leads to its proteolysis. The loss of *AKAP1* causes a decrease in PKA-dependent BAD and Drp1 phosphorylation, triggering their association with Bcl2 and FIS1, respectively, which enhances mitochondrial fission, mitochondrial ROS production, oxidative stress, cardiomyocyte death and myocardial dysfunction [151,173] (Table 2).

*AKAP5* (i.e., AKAP79 or AKAP75, AKAP150) is a PKA type II-specific anchoring protein for which roles and functions have been extensively investigated over the past years in cardiac physiological and pathophysiological conditions (Table 2). The *AKAP5* N-terminus exhibits a membrane targeting domain distributing the AKAP to the T-tubule network or sarcolemma inner leaflets, which facilitates its interaction with several proteins (e.g., AC5 and 6, PKC, F-actin, cadherin) (Table 2) [152,153]. In cardiomyocytes, *AKAP5*-anchored PKA mediates direct AC5 and AC6 phosphorylation to inhibit AC activity and cAMP production (Table 2; Figure 2B) [153,175]. *AKAP5* displays MAGUK and CaN binding domains in its central core. In the C-terminus, the AKB domain resides together with a leucine zipper motif that interacts with LTCC [82]. It has been reported that *AKAP5* brings PKA in proximity to LTCC, caveolin-3 and β-AR [154]. This supramolecular signaling complex regulates the PKA-induced sympathetic signaling activation and LTCC-mediated Ca^2+^ entry in cardiomyocytes [82,154] (Table 2; Figure 2B). In chronic MI, *AKAP5* KO mice develop exacerbated ventricular dilation, severe cardiac remodeling (i.e., fibrosis and large infarct), myocyte apoptosis and an enhanced hypertrophy with pulmonary edema [155]. The loss of cardiac *AKAP5* leads to impaired Ca^2+^ signaling, a defect in CICR (i.e., Ca^2+^ transient and SR load), a diminution in cardiac Ca^2+^ regulatory protein phosphorylation and contractility deficiency (Table 2) [155]. In addition, a decrease in the cardiac *AKAP5* expression level under pathological stress has been reported [155]. These data highlight *AKAP5* cardioprotective functions. However, this is challenged by another study showing that *AKAP5*-anchored CaN participates in NFATc3 activation, which down-regulates the K_v_ channel expression level and reduces *I*_Kv_ post-MI. This prolongs the action potential duration and favors arrhythmia susceptibility (Table 2; Figure 2B) [156].

*AKAP6* (i.e., mAKAP or muscle AKAP, AKAP100) is a PKA type II-specific anchoring protein which, by interaction with nesprin-1, confers a specific localization to the outer nuclear membrane (Table 2) [157]. *AKAP6* has also been reported to distribute, to a lesser extent, to the SR [176]. However, this SR distribution has been challenged and it has been proposed instead that this could reflect the close proximity between the nuclear envelope and cardiac dyads (i.e., T-tubule and SR). In the heart, *AKAP6*, in addition to binding PKA type II and nesprin-1α, clusters a large number of proteins either constitutively or temporarily including PLCε (phospholipase Cε), AC5, Epac1, PDE4D3, RyR2, NCX, HIF1-α Hypoxia Inducible Factor 1α), MEF2 (myocyte enhancer factor-2), kinases (PKCε, PKD, MEK5, ERK5, PDK1, RSK3) and phosphatases (CaN and PP2A) (Table 2) [158]. The *AKAP6*-PKA-PDE4D3-AC5 complex negatively regulates the AC5-dependent cAMP production as a feedback loop (Figure 2C). Activated AC5 produces a pool of cAMP that mobilizes *AKAP6*-anchored PKA. Thereafter, the kinase phosphorylates both AC5 and the *AKAP6*-anchored PDE4D3 that, respectively, inhibit AC5-dependent cAMP production and trigger local cAMP degradation by PDE4D3 (Table 2; Figure 2C) [159]. Furthermore, *AKAP6* regulates cellular oxygen homeostasis by modulating the expression level of the transcription factor HIF1-α [160] (Table 2). HIF1-α accumulation has been reported as an early MI marker [161]. Although *AKAP6* in normoxia mediates HIF1-α ubiquitination and degradation, hypoxia inhibits this process and leads to HIF1-α accumulation (Figure 2C). This increases HIF1-α/HIF1-β complex formation, which initiates transcription of pro-survival genes (e.g., pro-angiogenic, metabolic and anti-apoptotic) and cell survival under ischemic stress (Table 2; Figure 2C) [160].

*AKAP8* (i.e., AKAP95) is a PKA type II-specific anchoring protein highly expressed in the heart and anchors PKA RIIα with higher affinity than RIIβ (Table 2) [162,163]. *AKAP8* associates with the nuclear matrix during interphase and mediates chromosome condensation at mitosis [177]. However, the role of *AKAP8* remains elusive in adult cardiomyocytes that do not divide under physiological conditions. Post-MI, microRNA-21 (miR-21) is upregulated at the border of infarction zones, which exhibits cardioprotective functions against I/R-induced apoptosis [178]. Recently, it has been reported that miR-21 targets and reduces the *AKAP8* expression level, leading to cardiomyocyte protection against apoptosis induced by I/R events (Table 2) [164]. However, the molecular mechanism underlying the inhibition of *AKAP8* that mediates anti-apoptotic effects during I/R events needs to be further investigated.

*AKAP10* (i.e., D-AKAP2) is a dual-specificity AKAP that mainly associates with mitochondria but is found, to a lesser extent, in the cytoplasm or at the plasma membrane (Table 2; Figure 2D). In addition to the AKB, the *AKAP10* amino acid sequence includes two RGS (Regulators of G protein Signaling) homology domains that interact with the small GTPases Rab4 and Rab11 (Table 2; Figure 2D). Furthermore, *AKAP10* also displays a PDZ (PSD-95/DlgA/ZO-1) binding motif, which docks NHERF (Na^+^/H^+^ Exchanger Regulatory Factor) isoforms and ensures the connection between the anchoring protein and the solute carrier [148,179]. *AKAP10* exhibits a functional single-nucleotide polymorphism (SNP) in which the Ile in position 646 is substituted by a Val [165]. This polymorphism affects the AKB domain by increasing the affinity of *AKAP10* for type I PKA anchoring without affecting the one of type II. Patients with this functional *AKAP10* SNP have an increase in HR with a low HR variability predisposing to sudden cardiac death [180]. This is replicated in a mice model mutated in the *AKAP10*-AKB domain that displays modifications in the HR and baroreceptor function with sinoatrial and atrioventricular blocks and arrhythmias (i.e., extended P-P and P-R intervals on ECGs) [180]. These findings suggest a role for *AKAP10* in the regulation of the heart rhythm. However, the underlying molecular mechanisms need to be further investigated. Interestingly, patients with a functional *AKAP10* SNP without hypercholesterolemia have an increased prevalence for MI, which implies the I646V nucleotide substitution as a cardiac ischemic risk factor (Table 2) [166].

*AKAP12* (i.e., gravin or AKAP250, SSeCKS) is a PKA type II-specific anchoring protein that locates in the cytoplasm or associates with the cytoskeleton network and plasma membrane (Table 2; Figure 2D). The *AKAP12* N-terminus exhibits a myristoylation site and a MARCKS effector homology domain favoring the docking to the plasmalemma [167]. In the heart, *AKAP12* scaffolds multiple signaling proteins including PKA type II, PKC, PLK1 (Polo-Line Kinase-1), PDE4D, CaN and β-Ars (β-Arrestins) (Table 2) [168,169]. *AKAP12* binds to the β2-AR C-terminus and mediates phosphorylation of the GPCR by PKA. This triggers desensitization/resensitization cycling of the receptor (Table 2; Figure 2D) [168,169]. *AKAP12*-mutated mice without the ability to anchor both PKA and β2-AR exhibit a diminution in PKA-dependent β2-AR phosphorylation. This leads to the reduction in β2-AR desensitization and increases cardiac function and contractility [170]. Post-MI, cardiac Ang II levels increase which trigger cardiac remodeling and apoptosis. It has been reported that Ang-II infusion in *AKAP12* KO mice mediates exacerbation of TGF-β1 signaling and Smad2/3 phosphorylation, leading to cardiac oxidative stress, apoptosis and fibrosis (Table 2; Figure 2D) [171]. In contrast, *AKAP12* overexpression suppresses the Ang-II-induced TGF-β1 signaling activation and fibrosis, supporting the cardioprotective properties of *AKAP12* (Table 2; Figure 2D). Of note, the cardioprotective function of *AKAP12* in cardiac fibroblasts has been previously reviewed [172].

### 5.3. Miscellaneous Cardiac AKAPs

*AKAP7* (i.e., AKAP15 or AKAP18) is a PKA type II-specific anchoring protein with several isoforms which have been reported to be present in the heart including shorter (i.e., *AKAP7*α, *AKAP7*β) and longer variants (i.e., AKAP7γ/δ (respectively, for human/rat)). All isoforms present a common C-terminus that includes the AKB domain. Differences in *AKAP7* isoforms occur in the length of the N-terminus. The distribution at the plasma membrane of shorter *AKAP7* isoforms is facilitated by palmitoylation and myristoylation of N-terminal residues. The leucine zipper motif present in the C-terminus mediates *AKAP7*α interaction with LTCC [181,182]. Upon β-AR stimulation, *AKAP7*α-anchored PKA mediates LTCC phosphorylation on α and β subunits to potentiate *I*_CaL_ and cardiac contractility [183,184].

AKAP7γ/δ localizes at the SR membrane of cardiomyocytes and interacts with phospholamban (PLB) [185]. Upon β-AR stimulation, *AKAP7*γ/δ-anchored PKA phosphorylates PLB at Ser16. This phosphorylation alleviates specific PLB inhibition of SERCA2, which enhances SERCA2 ATPase activity, Ca^2+^ uptake in the SR and cardiac relaxation [185]. In addition, anchored PKA phosphorylates PDE3A1 bound to SERCA2, which hydrolyzes the pool of cAMP in the vicinity of the AKAP7γ/δ-PKA-PLB-SERCA2 complex to temporally regulate the PKA-induced SR Ca^2+^ uptake [186]. Furthermore, by anchoring Protein Phosphatase Inhibitor-1 (I-1), *AKAP7*γ/δ facilitates its phosphorylation by PKA at Thr35, which inhibits PP1 activity and the dephosphorylation of PLB by this phosphatase [187]. In contrast to the shorter isoforms, AKAP7γ/δ scaffolds a supramolecular signalosome complex regulating cardiac relaxation in space and time.

*AKAP9* (i.e., AKAP450, AKAP350, Yotiao, CG-NAP, Hyperion) is a PKA type II-specific anchoring protein that scaffolds PDE4D and AC9 and exhibits in its C-terminus a PP1 binding region and the AKB domain. *AKAP9* distributes in the cytoplasm or is found associated with the cytoskeleton network, at the plasmalemma and at the Golgi apparatus. Yotiao, which is the shortest *AKAP9* isoform, interacts with KCNQ1 and its C-terminus leucine zipper motif. In cardiomyocytes, KCNQ1 (i.e., α-subunit) and KCNE1 (i.e., accessory β-subunit) form the potassium channel and trigger the slow delayed rectifier K^+^ current (*I*_Ks_), which regulates cardiac action potential late-phase repolarization [188]. Upon β-AR stimulation, Yotiao-anchored PKA phosphorylates KCNQ1 at Ser27, increasing *I*_Ks_, which shortens the action potential duration to maintain diastolic time intervals [188]. By anchoring PP1 and PDE4D in the vicinity of KCNQ1, Yotiao mediates spatiotemporal regulation of *I*_Ks_ by β-AR activation. Patients with mutations in the KCNQ1 and Yotiao interaction regions (e.g., G589D or Q1570L, respectively) exhibit long QT syndrome [189]. These mutations abolish the KCNQ1–Yotiao association, increasing the distancing between the potassium channel and Yotiao-PKA complex. This compromises PKA-induced KCNQ1 phosphorylation upon β-AR stimulation, which alters *I*_Ks_ regulation by the kinase, prolongs the action potential duration and delays cardiomyocyte repolarization. As a consequence, this extends the QT interval and favors the occurrence of fatal arrhythmias [189].

*AKAP13* (i.e., AKAP-lymphoid blast crisis (AKAP-Lbc) or Ht31) is a PKA type II-specific anchoring protein expressed in the cytoplasm of cardiac fibroblasts (CF) and cardiomyocytes [190,191]. In addition to PKA type II anchoring, *AKAP13* recruits additional signaling molecules including PKCη, PKD1, PDE4D, PKNα, MAPKs (i.e., MLTK, MKK3, p38α) and phosphatases (e.g., shp2) [192]. Among all known AKAPs, *AKAP13* exhibits a unique feature with the capacity to activate small Rho-GTPAses (i.e., RhoA and RhoC) through a specific guanine nucleotide exchange factor (GEF) domain [190,193]. This domain is composed of a tandem of Dbl-homology (DH) and plekstrin-homology (PH) motifs located in the central core of *AKAP13* [193]. The *AKAP13* expression level rises in response to cardiac stress and acts as an early modulator of compensatory cardiac hypertrophy [194]. In cardiomyocytes, mechanical stress or α1-AR stimulation activates RhoA by the *AKAP13* Rho-GEF domain, which, in turn, enhances the MAPK p38α signaling cascade (i.e., PKNα-MLTK-MKK3-p38α) [195,196]. Likewise, mechanical stress, α1-AR or AT1-R stimulation mobilizes *AKAP13*-anchored PKCη, which, in turn, phosphorylates and activates PKD1. This phosphorylation favors PKD1 dissociation from *AKAP13*, which consequently triggers MEF2-induced hypertrophic gene transcription and inhibition of cardiomyocyte apoptosis [197,198].

In CF, AngII stimulates AT1-R and activates the *AKAP13* Rho-GEF domain that eases myofibroblast differentiation with the induction of collagen, α-SMA expression and profibrotic gene transcription [191].

Additional AKAPs are known to be expressed in the heart including *AKAP11* (i.e., AKAP220), ezrin, SKIP, smAKAP and BIG2 [149,150,199,200,201]. However, their cardiac functions as PKA-anchoring proteins remain unexplored. Finally, several AKAPs have been identified in the vicinity of contractile myofilaments including synemin and myosprin, which are anchored at Z-lines, or cardiac troponin-T (cTnT) associated with thin filaments [202,203,204]. Although these AKAPs optimally regulate the contractile function by PKA, the composition of the respective protein macrocomplexes needs to be further investigated.

## 6. Phosphodiesterases (PDEs)

### 6.1. Structure and Function

The regulation of local cyclic nucleotide 3′-5′ monophosphate (cNMP; i.e., cAMP and cGMP) nanodomains in cardiomyocytes is conferred by a superfamily of enzymes that includes over 100 different PDE isoforms [205,206]. PDE ensures second messenger signaling termination by specifically hydrolyzing the cNMP phosphodiester bond to form inactive 5′-NMP (e.g., 5′-AMP or 5′-GMP). PDE isoforms result from various translation initiation sites or after alternative mRNA splicing. The PDE nomenclature includes a family number (i.e., 1–11), followed by a gene letter (i.e., A, B, C and D) and potentially a splice variant identification number. Among mammalian PDE families, three selectively hydrolyze cAMP (i.e., PDE4, 7 and 8), three are selective for cGMP (PDE5, 6 and 9) and five families hydrolyze both cyclic nucleotides with distinct properties (PDE1, 2, 3, 10 and 11) [3,206]. With the exception of PDE6, PDE7 and PDE11, all other PDE isoforms are expressed in the heart [31,207]. PDEs exhibit a conserved C-terminal catalytic domain, whereas differences occur in the N-terminal regulatory region that may include an enzyme dimerization domain, phosphorylation sites and binding sites to signalosome macrocomplexes (Figure 3A). The regulatory domain confers PDEs’ specificity. Although PKA may interact directly with some PDEs, their proximity to signaling molecules (i.e., kinases, phosphatases and AC) is generally insured by anchoring to AKAPs, which participates in intracellular signaling compartmentation. [208].

### 6.2. Cardiac PDEs with Physiological Function in MI

PDE1 isoforms are encoded by three different genes (i.e., *PDE1A*, *PDE1B* and *PDE1C*). Among cardiac PDE1 isoforms, PDE1A expression is upregulated in diseased hearts, PDE1B expression remains low and PDE1C, which represents the major PDE1 isoform, displays a striated expression pattern [209]. PDE1 exhibits two Ca^2+^/CaM binding domains positioned at the N-terminus region. Ca^2+^/CaM binding induces PDE1 conformational change, which enhances PDE hydrolytic activity [210]. In contrast, PKA- and CaMKII-dependent PDE1 phosphorylation reduces Ca^2+^/CaM binding and phosphodiesterase activity [209,211]. In ischemic cardiomyopathy, PDE1A expression increases in cardiomyocytes and in α-SMA (α-Smooth Muscle Actine)-positive myofibroblasts of the infarct border zone, but the expression level remains low in normal fibroblasts [212]. In myofibroblasts, PDE1A upregulation inhibits cAMP-Epac1-Rap1 and cGMP-PKG signaling pathways, leading to fibroblast transformation, extracellular matrix synthesis and collagen deposition [212].

PDE2A isoforms (i.e., PDE2A1, PDE2A2 and PDE2A3) are encoded by a unique *PDE2A* gene. The cAMP hydrolytic activity is controlled by cGMP binding to a tandem of regulatory GAF sites located in the PDE2A N-terminus region (Figure 3A) [213]. The PDE2A protein expression level upregulates in ischemic cardiomyopathy [214]. Interestingly, PDE2A overexpression in primary cardiac fibroblast culture leads to the degradation of cAMP and to an increase in α-SMA and CTGF (Connective Tissue Growth Factor) expression, which induce myofibroblast conversion and fibrosis [215].

Although it is clear that cardiac PDE2A compartmentalization modulates cardiac function, its role in cardiomyocytes remains controversial. In cardiac myocytes, the inhibition of mitochondrial PDE2A leads to a local accumulation of cAMP, which increases PKA-dependent phosphorylation of Drp1 and improves cell survival [216]. Furthermore, PDE2A exhibits cardioprotective effects by regulating the HR under chronic β-AR stress conditions [217]. Pharmacological inhibition of PDE2A activity increases the HR in WT mice, whereas its cardiac overexpression does the opposite in transgenic mice. The latter is explained by the cardiac PDE2A overexpression that mediates the reduction in β-AR/cAMP signaling and diminution of LTCC activity in ventricular cardiomyocytes and SAN (i.e., calcium clock) [217]. Post-MI, cardiac PDE2A overexpression blunts arrhythmia susceptibility and maintains proper cardiac contractile function under acute β-AR stimulation [217]. Additional studies performed under healthy conditions reported that the stimulation of β_3_-AR located in the T-tubule network induces NO synthesis by eNOS, which, in turn, triggers cGMP production [218]. This neo-synthetized cGMP activates a pool of PDE2A located in the T-tubule caveolin-rich region, which locally hydrolyzes the pool of cAMP produced after β_1_- and β_2_-AR stimulation and attenuates the inotropic response [218]. In chronic MI, the T-tubule network is disrupted and β_3_-AR redistributes at the cardiomyocyte plasmalemma [219]. This disorganizes β_3_-AR/cGMP/PDE2A signaling and consequently abolishes the PDE2A regulation on T-tubule cAMP levels [219].

In contrast to the regulation of PDE2, cGMP binding to PDE3 inhibits the cAMP hydrolysis activity. The PDE3 family is encoded by two different genes (i.e., *PDE3A* and *PDE3B*), which generate all isoforms (i.e., *PDE3A1, PDE3A2, PDE3A3* and *PDE3B*) (Figure 3A) [163,220,221]. Although PDE3A and PDE3B are both expressed in cardiomyocytes, PDE3A isoforms are more abundant [222]. The phosphodiesterase catalytic domain locates at the C-terminus of all PDE3 variants. The longest PDE3 isoforms (i.e., PDE3A1 and PDE3B) exhibit several phosphorylation sites (e.g., PKA, PKB and PKC) and hydrophobic loops favoring PDE3 insertion into lipid membranes at the N-terminal region (Figure 3A) [223]. PDE3A1 locates mainly at the SR subset, while the other PDE3A isoforms distribute in the cytoplasm. The PDE3B isoform is found in proximity to mitochondria at cardiomyocyte Z-bands and T-tubules (Figure 3B) [224]. Under the physiological condition of β-AR stimulation, PDE3 tightly regulates cardiac contractile function by temporally limiting chronotropic, inotropic and lusitropic effects. These effects are mediated after PDE3-dependent cAMP degradation that restricts PKA activity and reduces PLB and TnI phosphorylation (Figure 3B) [225]. PDE3A KO mice exhibit enhanced basal contractility due to an accumulation of cardiac cAMP, which increases Ca^2+^ transient amplitudes, SR Ca^2+^ load, SERCA2a activity and PKA-dependent PLB phosphorylation. Furthermore, the AKAP18/PKA/SERCA2a/PLB/PDE3A supramolecular complex located at the SR subset mediates the spatiotemporal control of basal cardiac contractility in an LTCC-independent fashion (Figure 3B) [226]. In mice, PI3Kγ orchestrates a cardiac signaling macrocomplex with PKA and PDE3B that regulates local cAMP signaling in space and time [227,228]. However, PDE3B KO mice present normal cardiac function [229]. In comparison to PDE3A, this points to an auxiliary regulation role for PDE3B in the heart. It is admitted that all PDE3 isoforms exhibit cardioprotective functions in a rodent model with I/R injury. PDE3A1 transgenic mice with MI triggered post-I/R injuries present reductions in infarct size and myocyte apoptosis. Furthermore, these mice also display a conserved contractile function with a preserved ejection fraction post-MI [225]. Inhibition of ICER (Inducible cAMP Early Repressor) and Bcl2 expression mediates these cardioprotective effects [225]. PDE3B KO mice present in vitro and in vivo cardiac protection (e.g., reduced infarct size and a preserved cardiac function) after acute MI in a cAMP-induced preconditioning process [224]. It has been proposed that PDE3B knockout mediates cardioprotective effects by favoring a local pool of cAMP-PKA signaling, which mediates mitoK_ca_ channel activation, ICEF (Ischemia-Induced Caveolin-3–Enriched Fractions) signalosome assembly, the decrease in ROS production and resistance to Ca^2+^-induced mPTP pore opening [224].

Of note, pan-PDE3 inhibitors (e.g., amrinone, milrinone) have been clinically used to treat heart failure patients [230]. Despite the short-term symptomatic improvement in heart function, long-term treatment leads to an increase in patient mortality due to sudden cardiac death [231,232]. Considering the roles of cardiac PDE3A vs. PDE3B, isoform-specific PDE3 inhibitors could pave the way to diversification of the MI therapeutic scheme.

Among PDEs, cAMP-specific PDE4s are by far the largest PDE family encompassing more than 20 different isoforms encoded by four distinct genes (i.e., PDE4A, PDE4B, PDE4C and PDE4D) that generate all alternatively spliced variants. These alternative splicings provide PDE4s with different sizes and molecular masses that are classified into long, short, super-short and dead-short isoforms. PDE4s exhibit a unique N-terminal domain (TD), a central region containing upstream conserved regions 1 and/or 2 (UCR1 and UCR2) and a highly conserved C-terminal catalytic domain. The N-terminal domain confers to PDE4s a specific subcellular location, binding to signalosome complexes and the modulation of enzymatic activity (e.g., kinase phosphorylation) (Figure 3A) [233]. PDE4A, PDE4B and PDE4D isoforms are expressed in the cardiovascular system [234,235]. Although PDE4s have limited effects in basal cardiac function, they modulate the HR and cardiac contractility in response to β-AR stimulation [233,236]. In mice, a fraction of cardiac PDE4B is recruited to the LTCC/AKAP18/PKA signalosome in the T-tubule network controlling LTCC under β-AR stimulation [235]. Similarly to PDE3A, cardiac PDE4D regulates SERCA2a activity and SR Ca^2+^ load and leak (Figure 3B) [235,237,238]. This redundancy between both PDE subtypes highlights the necessity to tightly control the cAMP concentration in proximity to the SR to ensure a proper regulation of calcium re-uptake and relaxation. Furthermore, it has been reported that AKAP9 orchestrates a supramolecular signaling complex encompassing PKA, PP1 and PDE4D at the plasmalemma in the vicinity of the cardiac potassium channel (see Section 5.3). PDE integration in this complex modulates, in time, the cAMP signaling that controls the cardiac action potential repolarization current [239]. PDE4D plays also a central role in β-Ars’ desensitization process by directly interacting with β-arrestin [240].

PDE4D KO mice exhibit an increase in the cardiac cAMP level at the Z-line, which triggers PKA-dependent RyR2 hyperphosphorylation and calcium leak [233]. Interestingly, the crossing of these PDE4D KO mice with mice harboring a mutation in the RyR2 PKA-dependent phosphorylation site (RyR2-S2808A) produces offspring that show protection against MI-induced sudden cardiac death [241]. This indicates a key role of PDE4D in the regulation of the dyadic cAMP concentration, maintaining a proper level of RyR2 PKA-dependent phosphorylation and activity (Figure 3B).

Finally, the PKA-dependent phosphorylation of Hsp20 (at Ser16) has been reported in rats to provide cardioprotection and to reduce myocardial apoptosis after I/R injury [242,243,244,245]. PDE4D directly interacts with Hsp20 and finely tunes its phosphorylation by locally modulating the cAMP level and PKA activity [246]. A protein–protein interaction (PPI)-disrupting peptide displacing PDE4D from Hsp20 binding in neonatal rat ventricular cardiomyocytes increases PKA-dependent Hsp20 phosphorylation and shows cardioprotective functions. These PPIs could be of interest as new therapeutic agents in MI [247].

### 6.3. Miscellaneous Cardiac PDEs

In addition to the major types, other PDEs hydrolyzing cAMP co-exist in the heart but exhibit lower expression (i.e., PDE8A and PDE10). Their roles in the regulation of cardiac physiology or post-MI remain unclear and need to be further characterized. Beside its detailed role in CF, PDE1 also modulates the cAMP-PKA signaling pathway in the SAN and regulates pacemaker activity [248]. PDE8A regulates the calcium current and SR leak in cardiomyocytes [249]. However, the molecular mechanisms underlying this regulation need to be further investigated. The dual-activity PDE10 has recently been described to co-distribute in CF and cardiomyocytes. In response to pathological stimuli, PDE10 triggers cardiomyocyte hypertrophy, myofibroblast transformation and cardiac remodeling [207].

## 7. Discussion

Cyclic AMP signaling is one of the best characterized signaling pathways and critically involved in many cellular and physiological processes. It is now widely accepted that in the heart, cAMP plays a central role in the regulation of cardiac function. This includes the binding of an extracellular ligand (i.e., hormones and neurotransmitters) to GPCR, which, via G proteins and AC, leads to an intracellular increase in cAMP content. Next, this triggers activation of intracellular downstream effectors (i.e., PKA, Epac, CNCCs, POPDC), which, in a synergistic process, mediate the adaptation of cardiomyocytes to distinct stimuli. The temporal regulation of the cAMP signaling depends mainly on AC and cAMP-PDE activities. AKAPs provide the means to achieve both spatial and temporal regulation of PKA signaling. By scaffolding signaling effectors and modulators in the vicinity of substrates, AKAPs ensure spatial specificity in signal transduction. In addition, they also facilitate temporal regulation of the cAMP signaling pathway by placing the kinase optimally vs. the cAMP nanodomains. Furthermore, by assembling multiprotein signal complexes including phosphatases and PDEs, AKAPs facilitate discrete temporal control of the cAMP signalosome [4].

Acute β-AR signaling is triggered to compensate MI-induced loss of cardiac pump efficacy. This process increases myocardial cAMP, which favors positive cardiac inotropic effects and left ventricular function. Historically, considering these beneficial effects of acute β-AR signaling, the use of β-adrenergic agonists or PDE inhibitors has been speculated to be beneficial for patients with MI and heart failure [231,250]. Despite the short-term symptomatic improvement in heart function, long-term treatment with β-AR signaling activators leads to an increase in patient mortality due to sudden cardiac death [231,250]. While catecholamines and myocardial cAMP initially help the stressed heart, chronically at high levels, they promote adverse cardiac remodeling, cardiac myocyte death, fibrosis replacement and progressive deterioration of cardiac function (i.e., the vicious circle of heart failure) [19,251]. In the same period, several studies reported the deleterious effects of increased cardiac cAMP appearing post-acute MI that causes myocardial perfusion–contraction mismatching, increases infract size and promotes ventricular arrhythmias [252,253]. Therefore, various therapeutic strategies have been envisioned among which β-blocker adjunction evokes remarkable beneficial effects and reduces mortality [254]. Several β-blocker classes have been developed with specific physical and biochemical properties, which are differentially recommended in the treatment of distinct cardiac pathologies (e.g., non-selective β-blockers and cardio-selective β1-blockers with or without intrinsic sympathomimetic activity (ISA)). Post-MI, the beneficial cardiac effects of β-blockers are primarily mediated by their negative chronotropic and inotropic features, which decrease myocardial oxygen demands, increase coronary blood flow (by extending the diastolic filling time) and favor perfusion in ischemic regions [255]. Despite the fact that β-blockers are generally considered well tolerable, they may cause severe side effects, leading to low compliance in patients. These effects are multiple and affect cardiovascular functions as much as other physiological functions, leading to fatigue, depression, cold extremities and erectile dysfunction. Cardiovascular unwanted effects include bradycardia, AV block, hypotension and bronchoconstriction. Although β-blockers are administrated to patients post-MI or with hypertension or arrhythmias, they are contraindicated in patients with hypotension, bradycardia, Raynaud phenomena, severe pneumopathy, renal insufficiency or diabetes mellitus [16]. Despite current medications, no cure exists for MI and mortality remains high. Contraindications and side effects forced the scientific and medical community to consider therapeutic alternatives to β-blockers. In consequence, it was envisioned to target β-AR/cAMP signaling downstream effectors and/or modulators to increase the specificity and efficacy of therapeutics in cardiac pathologies. Therefore, inhibition of PKA has become of interest as a novel putative therapeutic target to counteract the β-AR activation signaling cascade observed [256]. Initially, small molecules exhibiting a PKA inhibition (i.e., H89 and KT-5720) were envisioned [243,257]. These two compounds antagonize ATP docking in the PKA C subunit nucleotide binding pocket. In vitro, these PKA inhibitors reveal promising effects by reducing cardiomyocyte death induced by chronic β-AR stimulation [257,258]. Unfortunately, these molecules present a deficit in specificity of PKA inhibition [259]. Consequently, alternative PKA inhibitor tools have been conceived and tested. PKI peptides are derivative analogs from the endogenous PKI (protein kinase-A inhibitor) protein, which encompass a PKA-specific inhibitory domain [259,260]. These peptides exhibit potent and highly specific PKA inhibition properties [259]. Interestingly, β-AR stimulation failed to induce cardiac hypertrophy, fibrosis, myocyte apoptosis and decreased cardiac function in transgenic mice specifically overexpressing in the heart PKI peptide [29]. In one study, the authors observed a superior cardioprotective potential with selective inhibition of PKA compared to β-blocker therapy after myocardial infarction [29]. Despite a substantial diminution of cardiac adaptation to exercise, PKI seems a promising path that could be developed in the design of alternative therapies to β-blockers post-MI [29,30,106].

Almost two decades ago, the overexpression of cardiac AC6 in mice revealed unexpected promising outcomes post-MI and for which a reduction in the mortality rate, a decrease in cardiac remodeling and protection of cardiac function were reported [87]. Precise mechanisms underlying cardioprotective outcomes remain uncertain. However, the authors suggested that the positive effects observed with respect to contractile function could be the consequence of an increase in the PLB phosphorylation level and SR calcium load [87]. Therefore, gene therapy directing specific cardiac AC6 overexpression could be a possible therapeutic solution post-MI.

Human clinical trials demonstrated in various pathologies benefit from small interfering RNAs mediating a specific gene knockdown (reviewed in [261]). However, targeting cAMP signaling modulators or regulators may be drastic as these are involved in the regulation of key processes regulating the function of cardiomyocytes. However, silencing PKA substrates triggering adverse cardiac remodeling post-MI could be a possible envisioned therapeutic.

An elegant alternative with specific PDE activators would increase myocardial cAMP hydrolysis in distinct cAMP nanodomains and thus inhibit deleterious PKA signaling post-MI. However, such molecules are still in development or under characterization and have not yet been tested under cardiac pathological conditions [262].

Finally, targeting protein–protein interactions (PPI) in macromolecular signaling complexes orchestrated by AKAPs with PPI disruptors (e.g., peptides and small molecules) offers new exciting therapeutic options [142]. Peptides are easy to synthesize, highly specific and exhibit strong affinity with minimal toxicity and immune responses. Several PPI disruptors have been designed to target interactions between AKAP/PKA or AKAP/PKA substrates and their efficacy evaluated in the heart (reviewed in [142]). As the PKA/AKAP interaction domain is structurally common to all complexes, PPI disruptors (e.g., Ht31, RIAD, SuperAKAP-*IS*) would lack specificity and should not be considered as potential therapeutics. Alternatively, PPI disruptors targeting the anchoring domain between AKAPs and PKA substrates demonstrate higher specificities and efficiencies. Among them, a PPI-disrupting peptide competing in the AKAP18δ/PLB interaction in cardiomyocytes alters PLB phosphorylation and calcium re-uptake [185]. In addition, a PPI-disrupting peptide targeting the mAKAP/CaN interaction decreases in vitro cardiac hypertrophy [263]. Finally, TAC mice overexpressing a peptide disrupting the AKAP-Lbc/p38 interaction exhibit in vivo a reduction in cardiac hypertrophy [196]. Therefore, PPI-disrupting peptides provide new exciting and specific therapeutics that can be used post-MI as an alternative to β-blocker therapy.

The beginning of this century has seen the emergence of innovative therapeutic approaches focusing on the engineering of allosteric modulators and “biased” ligands to selectively activate one of GPCR-associated intracellular signaling events (for review, see [264]). Therefore, the discovery of β-Ars-biased agonists has been expected in the cardiology field to selectively trigger cardioprotective β-Ars signaling over the β-AR Gα_s_ coupling. In this context, β2-AR-pepducin ICL1–9, a cell penetrating lipopeptides derived from the amino acid sequence of the first β2-AR intracellular loop, exhibits promising β-Ars-biased agonist properties. It selectively activates β2-AR-associated β-Ars signaling, internalizes β2-AR and promotes adult cardiomyocyte contractility [265,266]. In addition, intramyocardial injection of β2-AR-pepducin ICL1–9 during I/R injuries reduces infarct size, maintains cardiomyocyte survival, improves cardiac function and therefore may offer an elegant alternative to β-blocker treatment post-MI [267].

It is noteworthy that cAMP signaling displays ubiquitous distribution in all cells and organs over the body, and caution should be noted as the design of medications regulating such signaling could introduce side effects. To avoid adverse effects, new therapeutic molecules developed to treat cardiac diseases could, with benefit, be delivered efficiently and rapidly directly to the myocardium in a tissue-specific manner. Such a strategy would improve the treatment of cardiovascular diseases in general (i.e., CHD, hypertrophic cardiomyopathies, acute or chronic HF, rhythm disorders) and would abolish or minimize undesirable side effects. In addition, heart-targeting systems would support the development of new therapeutic agent classes, such as PPI disruptors (peptides, small compounds) or nucleic acids (siRNA) [268,269,270]. Alternatively, gene therapy strategies using associated adenoviruses (AAVs) exhibiting preferential cardiac tropism would specifically direct expression of cAMP modulators/regulators (e.g., PDE, AC), peptides (e.g., PKI, PPI disruptors) or siRNA in the heart and could be considered as therapeutic options [259,271].

In conclusion, although the management of patients post-MI is well established, strategies to improve disease outcome are still under investigation. Even if β-blockers remain the cornerstone therapy in CHD, they also exhibit unwanted effects and are inappropriate for some patients [272]. As described here, the cAMP signaling and its compartmentalization play a crucial role in cardiac physiology and are extensively modified post-MI. Therefore, new therapeutic options targeting cAMP signaling would deserve further investigations and might offer beneficial avenues to prevent fibrosis in CHD or susceptibility to arrhythmias, but also to reduce infarct size and/or to preserve cardiac contractile function.

## Figures and Tables

**Figure 1 cells-10-00922-f001:**
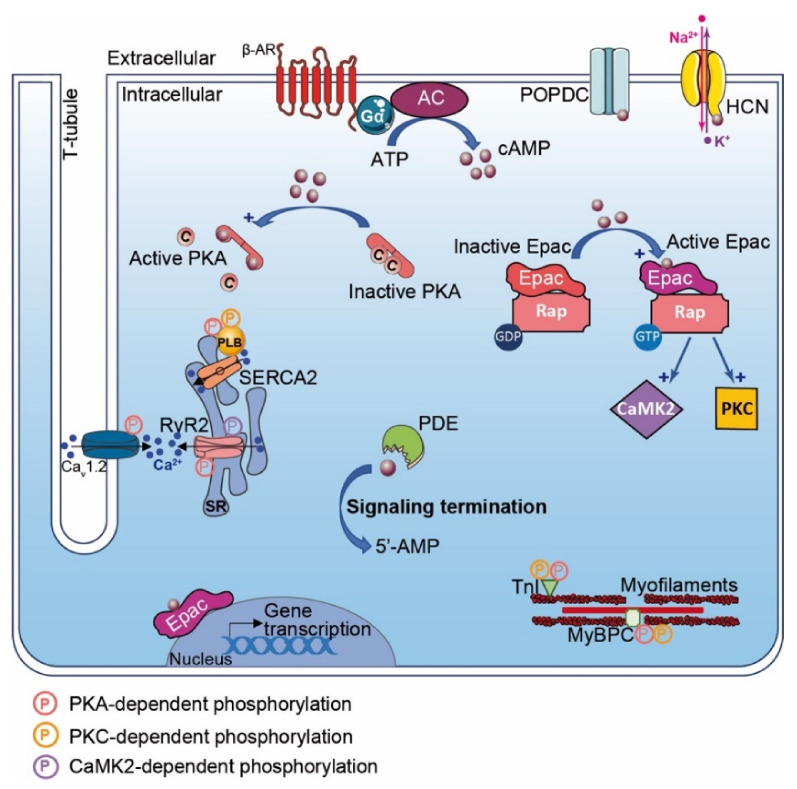
β-Adrenergic Receptor and cAMP signaling pathways in the heart. β-Adrenergic Receptor (β-AR) activates adenylyl cyclase and generates pools of cAMP. cAMP (dark red-filled circles) has effects on a range of down effectors encompassing: PKA, Epac, POPDC, hyperpolarization activated cyclic nucleotide (HCN) channel and phosphodiesterases (PDEs). PKA activation leads to phosphorylation (P in pink circles) of specific substrates regulating Ca^2+^ flux and cardiac excitation–contraction coupling (CEC) (e.g., PLB, Ca_V_1.2 (LTCC), RyR2, TnI, MyBPC). Cyclic AMP binding to Epac favors exchange of RAP-GDP into RAP-GTP, which activates phosphorylation by PKC and CaMK2 (P in yellow and purple circles, respectively). Activated Epac regulates gene transcription. Cyclic AMP binding to HCN channels triggers ion flux (Na^+^, K^+^) and hyperpolarization. Local concentration of cAMP gradient is limited by phosphodiesterases (PDEs), which hydrolyze cyclic nucleotide in inactive 5′-AMP, leading to termination of signaling.

**Figure 2 cells-10-00922-f002:**
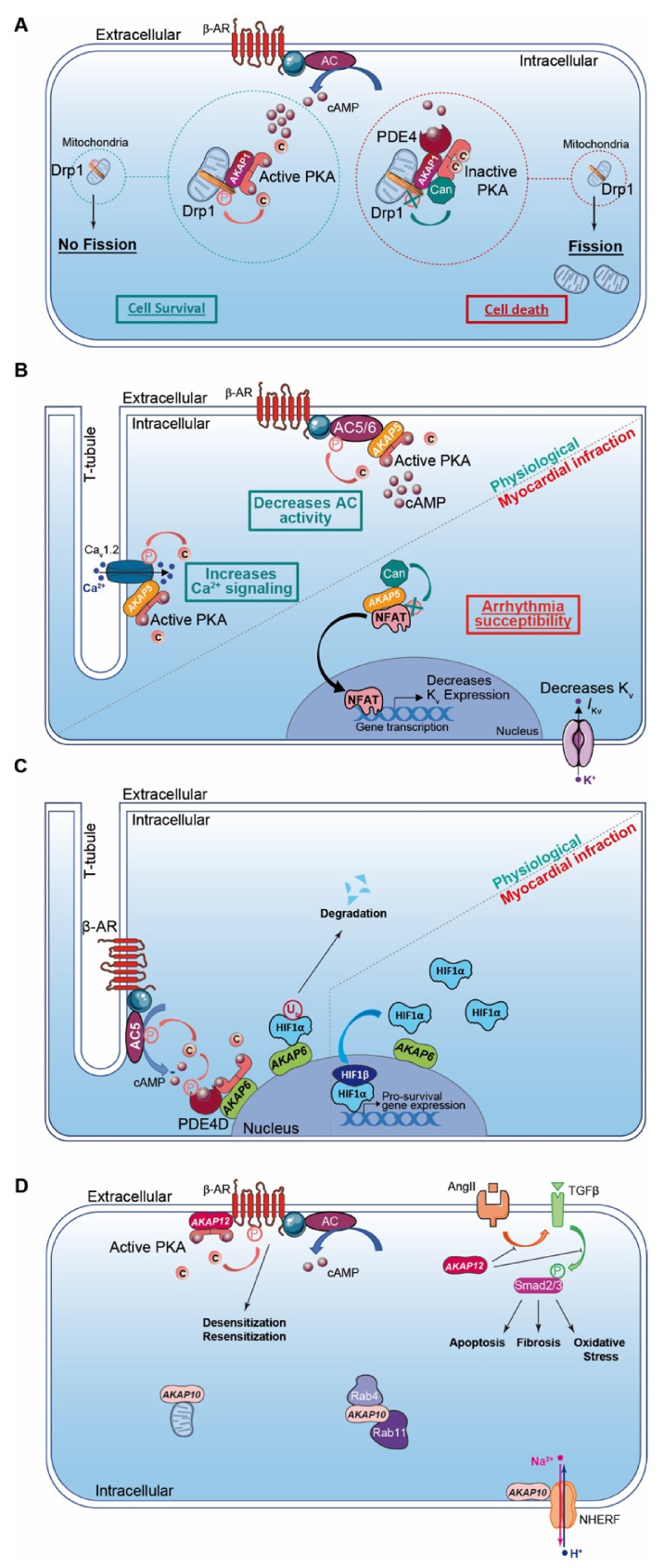
Cardiac AKAPs and cAMP signaling compartmentalization in myocardial infarction (MI). (**A**) AKAP1 coordinates at the mitochondria a cardioprotective macrocomplex that mediates phosphorylation of Drp1 by anchored PKA, which inhibits mitochondrial fission and leads to cell survival (left). This process is counterbalanced by CaN recruitment on AKAP1 signaling complex, which, in contrast, favors Drp1 dephosphorylation and mitochondrial fragmentation (right). (**B**) Role of cardiac AKAP5 under physiological conditions and after MI. In cardiomyocytes, AKAP5-anchored PKA mediates direct AC5 and AC6 phosphorylation to inhibit AC activity and cAMP production (top left). AKAP5 brings PKA in proximity to LTCC, which regulates Ca^2+^ entry (bottom left). AKAP5 anchors CaN and participates in NFATc3 activation, which down-regulates K_v_ channel expression level, reduces *I*_Kv_, prolongs action potential duration and favors arrhythmia susceptibility post-MI (right). (**C**) Role of cardiac AKAP6 under physiological conditions and after MI. Activated AC5 produces a pool of cAMP that mobilizes AKAP6-anchored PKA. PKA phosphorylates AC5 and AKAP6-anchored PDE4D that, respectively, inhibit AC5-dependent cAMP production and trigger local cAMP degradation by PDE4D (left). Under physiological conditions, AKAP6 mediates HIF1-α ubiquitination and degradation, while hypoxia inhibits this process and leads to HIF1-α accumulation. HIF1-α complexes with HIF1-β and initiates transcription of pro-survival genes to favor cell survival under ischemic stress (right). (**D**) Cardiac AKAP10 and AKAP12 under physiological conditions. In the heart, AKAP10 distributes to the mitochondria, in the cytoplasm (with small GTPases Rab4 and Rab11) and at the plasmalemma (associated with Na/H exchanger (NHERF)). AKAP12 favors β-AR phosphorylation and triggers GPCR desensitization/resensitization cycling. In the heart, angiotensin II (AngII) activates cardiac TGFβ1 pathways, which favors oxidative stress, apoptosis and fibrosis. AKAP12 inhibits deleterious AngII and the TGFβ1 pathway and exhibits cardioprotective properties. AKAP: A-kinase anchoring protein; HIF1α: Hypoxia Induced Factor-1α; CaN: calcineurin; PDE4: phosphodiesterase 4; AC5/6: adenylyl cyclase 5/6; β-AR: β-Adrenergic Receptor.

**Figure 3 cells-10-00922-f003:**
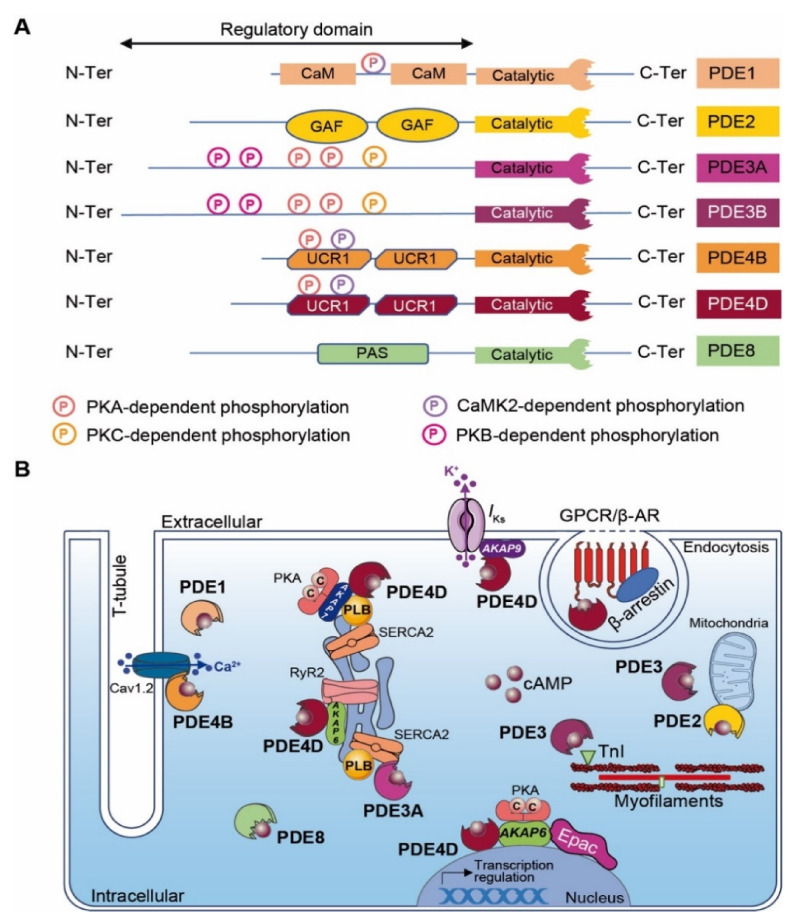
Cardiac cAMP-PDEs. (**A**) Structure of major cardiac cAMP-PDEs. PDEs exhibit a conserved C-terminal catalytic domain and a variable N-terminal regulatory domain. Kinase-dependent phosphorylation sites are indicated as circled P (light pink for PKA, yellow for PKC, purple for CaMK2, dark pink for PKB and mixed with light pink and purple for PKA and CaMK2). CaM: calmodulin binding domain; GAF: GAF (i.e., cGMP-dependent PDE, *Anabaena* adenylyl cyclases and E. Coli FhlA) domain; UCR: upstream conserved region; PAS: Per-Arnt-Sim domain; and PDE: phosphodiesterase. (**B**) Scheme of major cardiac cAMP-PDE compartmentalization. GPCR: G protein-coupled receptor; β-AR: β-Adrenergic Receptor; TnI: troponin I; RyR2: Ryanodine receptor-2; Epac: Exchange Protein Activated by cAMP, PKA: Protein Kinase-A; SERCA2: Sarco/Endoplasmic Reticulum Ca^2+^-ATPase; PLB: Phospholamban; AKAP: A-kinase anchoring protein.

**Table 1 cells-10-00922-t001:** Function of cardiac Gα_s_ and Gα_i_ protein-coupled receptors under physiological conditions and in myocardial infarction. α2-AR: alpha 2 adrenergic receptor; VT: ventricular tachycardia; β1-AR: beta 1 adrenergic receptor; β2-AR: beta 2 adrenergic receptor; β3-AR: beta 3 adrenergic receptor; M_2_R: muscarinic type 2 receptor; A_1_AR: A1 Adenosine Receptor; A_2A_AR: A2A Adenosine Receptor; A_2B_AR: A2B Adenosine Receptor; A_3_AR: A3 Adenosine receptor; GCCR: Glucagon Receptor; GLP1R: Glucagon Like Peptide 1 Receptor.

Receptors	Cardiac Function		In Myocardial Infarction	
**α2-AR** Alpha 2 adrenergic receptor	Coupled to Gα_i_ Inotropic - Chronotropic -	[33]	Prevents arrhythmias (VT)	[34]
**β1-AR** Beta 1 adrenergic receptor	Coupled to Gα_s_ Plasma membrane localization Inotropic + Chronotropic + Lusitropic +	[35]	Expression decreases Knockdown improves cardiac function	[36] [37]
**β2-AR** Beta 2 adrenergic receptor	Coupled to both Gα_s_ and Gα_i_ Restricted to T-tubules Inotropic + Chronotropic +/−	[35]	Redistributes to the plasma membrane Limits the infarct size Limits circulating TnI Reduces deleterious remodeling Restores cardiac function Partly inhibits inflammatory response	[37] [38] [39]
**β3-AR** Beta 3 adrenergic receptor	Coupled to both Gα_s_ and Gα_i_ Inotropic -	[40]	Expression increases Limits the infarct size Improves cardiac function Increases cell survival Reduces fibrosis	[41] [42]
**M_2_R** Muscarinic receptor type 2	Coupled to Gα_i_ Inotropic -	[35]	Upregulation in the remote zone Prevents arrythmia	[43]
**A_1_AR** A_1_A Adenosine Receptor	Coupled to Gα_i/o_ Depresses cAMP production	[44]	Decreases cell death Counteracts contractile dysfunction	[44] [45]
**A_2A_AR** A_2A_ Adenosine Receptor	Coupled to Gα_s_ Increases cAMP production	[44]	Decreases infarct size Improves cardiac contractility	[44] [45]
**A_2B_AR** A_2B_ Adenosine Receptor	Coupled to Gα_s_ Increases cAMP production	[44]	Cardioprotective properties	[44,45]
**A_3_AR** A3 Adenosine Receptor	Coupled to Gα_i/o_ Depresses cAMP production	[44]	Cardioprotective properties	[44,45]
**EP3** Prostaglandin EP3 Receptor	Coupled to Gα_i_ Depresses cAMP production	[35]	Upregulated Reduces infarct size	[46] [47]
**EP4** Prostaglandin EP4 Receptor	Coupled to Gα_s_ Increases cAMP	[35]	Upregulated Reduces fibrosis and hypertrophy Improves cardiac function	[46]
**GCCR** Glucagon Receptor	Coupled to Gα_s_ and Gα_i_ Inotropic +	[48]	Increases cell apoptosis Increases infarct size Inhibition improves cardiac function and decreases deleterious remodeling	[49]
**GLP1R** Glucagon Like Peptide 1 Receptor	Coupled to Gα_s_ Inotropic -	[13]	Reduces infarct size	[50,51]

**Table 2 cells-10-00922-t002:** Role of cardiac AKAPs in physiological and in myocardial infarction. AKAP: A-kinase anchoring protein; MI: myocardial infarction; PP1/2B/2A: Protein Phosphatase 1/2B/2A; PDE: phosphodiesterase; Drp1: Dynamin Related Protein 1; AC5/6: adenylyl cyclase 5/6; PKC: Protein Kinase C; LTCC: L Type Calcium channel; cAMP: cyclic Adenosine Monophosphate; Epac 1: Exchange Protein Activated by cAMP; RyR: Ryanodine Receptor; NCX: Na/Ca exchange; HIF1α: Hypoxia Induce Factor 1α MEF2: Myocyte Enhancer Factor 2; NHERF: Na^+^/H^+^ Exchanger Regulatory factor; CaN: calcineurin; β-Ars: β-Arrestins; β2-AR: β2-Adrenergic Receptor.

AKAP	Physiological Function		MI Alteration and Therapeutic Interest	
***AKAP1*** D-AKAP1 S-AKAP84 AKAP121 AKAP149	Dual AKAP Mitochondrial localization Anchors PP1, PP2B, PDE4, Drp1 Regulates mitochondrial dynamic Favors the phosphorylation of bad, prevents cell death	[140]	Inhibits mitochondrial fission, triggers cell survival Ubiquitination by Siah2 leads to ROS production, oxidative stress, mitochondrial dysfunction, cardiomyocytes death	[31] [151]
***AKAP5*** AKAP79 AKAP75 AKAP150	Type II AKAP T-tubule localization Anchors AC5 and 6, PKC, F actin, cadherin, LTCC Inhibits cAMP production Regulates LTCC mediating Ca^2+^ entry	[152] [153] [154]	*AKAP5KO* exhibits impaired Ca^2+^ signaling and cardiac dysfunction Reduces *I*K_v_ and favors arrythmia susceptibility	[155] [156]
***AKAP6*** mAKAP AKAP100	Type II AKAP Outer nuclear membrane Anchors nesprin1a, AC5, Epac1, PDE4D3, RyR2, NCX, HIF1a, MEF2, kinases, PP2A and PP2B Regulates cAMP production through AC5 and PDE4D3 Regulates oxygen homeostasis through HIF1a ubiquitination	[157] [158] [159] [160]	*AKAP1* inhibition induced ubiquitination of HIF1α and transcription of pro-survival genes	[160,161]
***AKAP8*** AKAP95	Dual AKAP (binds PKA RIIα >>> RIIβ	[162,163]	Specific AKAP8 inhibition decreases cell apoptosis	[164]
***AKAP10*** D-AKAP2	Dual AKAP Mitochondrial localization Anchors NHERF, Rab4 and Rab11	[165]	Polymorphism I646V increases susceptibility to MI	[166]
***AKAP12*** Gravin AKAP250 SSeCKS	Type II AKAP Plasma membrane, cytoskeleton or cytoplasm localization Anchors PKC, PLK1, PDE4D4, CaN, β-Ars, β2-AR β2-AR desensitization	[167] [168] [169] [170]	*AKAP12*KO develops apoptosis, fibrosis, oxidative stress in response to angiotensin II *AKAP12* overexpression suppresses fibrosis	[171] [172]

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
