# Peer review of "Cardiac cAMP-PKA Signaling Compartmentalization in Myocardial Infarction"

_cells, 2021, doi:10.3390/cells10040922_

Round 1

Reviewer 1 Report

In this manuscript, the alterations in the different components of cAMP signaling due to myocardial infarction (MI) are reviewed, focusing mainly on the PKA pathway, although considering other pathways, especially that of the Epac proteins. Furthermore, special emphasis is placed on the compartmentalization of the cAMP signal in cardiac myocytes and its regulation by AKAP proteins or phosphodiesterases. From the review, the selective action on specific components of the cAMP signaling pathway as a therapeutic alternative to beta-blockers in MI is assessed.

The subject is interesting, although complex, and the lack of concrete studies in many aspects of the compartmentalization of the cAMP route is well resolved, resulting in a complete and interesting review, both for specialists on the subject and for those who want to get started on it.

Some minor suggestions and comments:

- Page 12, 1st paragraph: the definition of the types of CNCC is a bit confusing, since the same definition is used for HCN and CNG.

- The AKAP 1,5 and 6 signal compartmentalization schemes are very interesting and make it easy for the reader to understand. However, similar schemes are not included for other AKAPs (8, 10, 12) whose function could also be altered in post-MI hearts. Is there a reason for this exclusion?

- Page 21: neither the structure nor the compartmentalization for PDE1, which also hydrolyzes cAMP in the heart and whose role in myocardial infarction is described in the text, is not included in Figure 3. If there is no reason for such exclusion, the authors should correct this.

- Pages 28-29: The sections "Author contribution", "Funding", "Institutional review board statement", etc. are not properly covered.

Author Response

Revision of ms. no. cells-1166000

Point-by-point response to the reviewers:

We would like to thank reviewers for their comments, which we have addressed in full as specified below, and which we think has significantly improved the review.

Specific comments (please note that line# references may have shifted somewhat in the revised text):

Reviewer #1:

In this manuscript, the alterations in the different components of cAMP signaling due to myocardial infarction (MI) are reviewed, focusing mainly on the PKA pathway, although considering other pathways, especially that of the Epac proteins. Furthermore, special emphasis is placed on the compartmentalization of the cAMP signal in cardiac myocytes and its regulation by AKAP proteins or phosphodiesterases. From the review, the selective action on specific components of the cAMP signaling pathway as a therapeutic alternative to beta-blockers in MI is assessed.

The subject is interesting, although complex, and the lack of concrete studies in many aspects of the compartmentalization of the cAMP route is well resolved, resulting in a complete and interesting review, both for specialists on the subject and for those who want to get started on it.

Author response: We appreciate that reviewer though our review complete and interesting for non-specialist as well as for specialists on the subject and agreed about his/her concerned regarding manuscript presentation.

Some minor suggestions and comments:

- Page 12, 1st paragraph: the definition of the types of CNCC is a bit confusing, since the same definition is used for HCN and CNG.

Author response: We agree with Reviewer’s comment about confusion around CNCC definition versus HCN and CNG. Therefore, we followed his/her advice and modified 1st paragraph corresponding to this description. (p14, l.459-467): “CNCC are a heterogeneous superfamily of ion channels activated by cAMP or cGMP (Figure 1). The heart expresses two subtypes of this superfamily, which include hyperpolarization-activated cyclic nucleotide-gated ion (i.e., HCN) and cyclic nucleotide-gated ion (i.e., CNG) channels. Both HCN and CNG channels assemble at the plasmalemma in tetramer complexes. The transmembrane core consists of alpha-helical segments that forms the ion-conducting pore, which in case of HCN channel exhibits a supplementary voltage sensor domain conferring specifically to this channel volt-age-dependent gating properties [133]. The C-terminal extremity of HCN and CNG contains a CNB domain that binds cAMP and cGMP.”

- The AKAP 1,5 and 6 signal compartmentalization schemes are very interesting and make it easy for the reader to understand. However, similar schemes are not included for other AKAPs (8, 10, 12) whose function could also be altered in post-MI hearts. Is there a reason for this exclusion?

Author response: To address this point, we though schemes about roles of AKAP8 and AKAP10 under physiological conditions versus MI were difficult to summarize and achieve. Moreover, AKAP8 function in heart remains elusive and therefore we decided to not draw a figure about it. Concerning the AKAP10, the representation of polymorphisms complicates the understanding of the figure. However, we added a new panel in figure 2 (Fig. 2D) about AKAP10 subcellular localization in cardiomyocytes under physiological condition. Furthermore, we also depicted cardiac AKAP12 function in this new panel of figure 2 (Fig. 2D).

- Page 21: neither the structure nor the compartmentalization for PDE1, which also hydrolyzes cAMP in the heart and whose role in myocardial infarction is described in the text, is not included in Figure 3. If there is no reason for such exclusion, the authors should correct this.

Author response: We thank the Reviewer for pointing this out. We agree with his/her comments and there is no particular reason for PDE1 exclusion from original figure 3. Therefore, we have now included PDE1 structure in Figure 3A and its compartmentalization in Figure 3B.

- Pages 28-29: The sections "Author contribution", "Funding", "Institutional review board statement", etc. are not properly covered.

Author response: We have now modified this section properly.

Author Contributions: Conceptualization, AS.C. and G.P.; writing—original draft preparation, AS.C.; writing—review and editing, G.P.; visualization, AS.C.; supervision, G.P.; funding acquisition, G.P. All authors have read and agreed to the published version of the manuscript.

Funding: This work has benefitted funds from Inserm (ASC and GP), ANR (ANR-20-CE18-0017-01 to GP) and LabEx Lermit fellowship (ANR-10-LABX-33 to ASC). UMR-S1180 is a member of the Laboratory of Excellence in Research on Medication and Innovative Therapeutics supported by the Agence Nationale de la Recherche (ANR-10-LABX-33) under the program “Investissements d’Avenir” (ANR-11-IDEX-0003-01).

Institutional Review Board Statement: Not applicable

Acknowledgments: We are grateful to members of Inserm UMR-S1180 and Kjetil Tasken (Institute of Cancer Research, Oslo University Hospital, Norway) for critical reading of the manuscript.

Conflicts of Interest: The authors declare no conflict of interest.

Reviewer 2 Report

Overall, this is a very comprehensive review of cAMP-PKA signaling and compartmentation in the heart.  In fact, it is one I plan on using extensively and having all new students in my lab read.  There are only minor things that would make the review a little better.

  1. It would be helpful to have a graphical abstract showing the basic schematic of Galphas and Galphai signaling to PKA in the beginning.
  2. In the introduction, a more detailed description on the importance of compartmentalization would be helpful.  There is really only a sentence or two about this and perhaps a history of its findings would be helpful.
  3. There is new exciting evidence about beta receptors found at internal locations in the cell.  This would be good to mention and speculate on
  4. Additionally, biasing receptors or using pepducins should be added and their use in heart disease can be speculated on.
  5. On Page 9, it would be good to speculate on the novel data that PKA holoenzyme does not come apart during activation.  How would this help compartmentation
  6. In section 4.2.2, there is a paragraph that is duplicated.  Only one of these is needed.

Author Response

Revision of ms. no. cells-1166000

Point-by-point response to the reviewers:

We would like to thank reviewers for their comments, which we have addressed in full as specified below, and which we think has significantly improved the review.

Specific comments (please note that line# references may have shifted somewhat in the revised text):

Reviewer #2:

Overall, this is a very comprehensive review of cAMP-PKA signaling and compartmentation in the heart.  In fact, it is one I plan on using extensively and having all new students in my lab read. There are only minor things that would make the review a little better.

Author response: We really appreciate Reviewer’s comment regarding our review.

It would be helpful to have a graphical abstract showing the basic schematic of Galphas and Galphai signaling to PKA in the beginning.

Author response: As suggest by Reviewer we generated a new graphical abstract that includes Galphas and Galphai signaling and a general overview of the cAMP signaling.

In the introduction, a more detailed description on the importance of compartmentalization would be helpful.  There is really only a sentence or two about this and perhaps a history of its findings would be helpful.

Author response: We agree with Reviewer, and therefore included an additional paragraph in the introduction about the discovery of cAMP compartmentalization in the heart. P2.l55-l71: ‘Evidence of cardiac cAMP signaling compartmentalization relies on experiments performed in late 1970s [6],[7],[8]. Heart perfused with isoproterenol (a b-AR agonist) raises the strength of contraction (i.e., inotropic response) and increases myocardial cAMP level. The latter mobilizes PKA, which in turn mediates cascade activation of the phosphorylase kinase and the glycogen phosphorylase triggering glycogen breakdown. Although PGE1 (agonist of prostaglandin E1 receptor) perfusion in heart increases cAMP production and PKA activity, it failed to regulate cardiac contractile function and glycogen metabolism [7],[9]. At the cellular level, adult rat ventricular myocytes stimulated with glucagon, glucagon like peptide-1 and b2-AR agonist exhibit for all an increase in intracellular cAMP content but distinct cellular responses that are specific to each stimulus. Activation of b2-AR enhances positive inotropic response, while the glucagon exerts positive inotropic and lusitropic (i.e., rate of myocardial relaxation) effects. In contrast, glucagon like peptide-1 ensures a modest negative inotropic response [10],[11],[12],[13]. Therefore, restraining the cardiac cAMP signaling to distinct intracellular compartments provides means to achieve specific cardiomyocytes adaptation to various extracellular stimuli. Therefore, the concept of cardiac cAMP signaling compartmentalization was established (for review see [14]).’

There is new exciting evidence about beta receptors found at internal locations in the cell.  This would be good to mention and speculate on

Author response: We have included a new paragraph in section 2.2.2: b- adrenergic receptors (b-AR) (p7 l158-165) about inside cell b-AR signaling located at the Golgi apparatus and endosomes. ‘The classic model of b-AR activation depicts initiation of the signaling from the plasmalemma; however, this has been challenged by recent findings reporting also pools of active b1-AR/Gas and b2-AR/Gas inside cell at the Golgi apparatus and endosomes [41],[42]. The endosome-localized b2-AR ensures specific transcriptional activity, while functions associated with b1-AR distributed at the Golgi apparatus remains un-explored [43]. These findings add an additional level of complexity to the cAMP signaling compartmentalization and should be further investigated under cardiac pathophysiological conditions, as they may offer new potential therapeutic targets.’

Additionally, biasing receptors or using pepducins should be added and their use in heart disease can be speculated on.

Author response: We Thank Reviewer for mentioning this really interesting point, and therefore we included an additional paragraph about ‘biased’ ligands and β2-AR-pepducin in the discussion section (p29, l1005-1016). ‘The beginning of this century has seen the emergence of innovative therapeutic approaches focusing on the engineering of allosteric modulators and ‘biased’ ligands to selectively activate one of GPCR-associated intracellular signaling (for review see [272]). Therefore, the discovery of β-Ars-biased agonists have been expected in the cardiology field to selectively trigger cardioprotective β-Ars signaling over the β-AR Gas coupling. In this context, β2-AR-pepducin ICL1-9, a cell penetrating lipopeptides derived from the amino acids sequence of the first β2-AR intracellular loop, exhibits promising β-Ars-biased agonist properties. It selectively activates β2-AR-associated β-Ars signaling, internalizes β2-AR and promotes adult cardiomyocytes contractility [273],[274]. In addition, intramyocardial injection of β2-AR-pepducin ICL1-9 during I/R injuries reduces infarct size, maintains cardiomyocytes survival, improves cardiac function, and therefore may offer an elegant alternative to β -blockers treatment post-MI [275].’

On Page 9, it would be good to speculate on the novel data that PKA holoenzyme does not come apart during activation.  How would this help compartmentation

Author response: To answer to this point, we have extended and reorganized PKA paragraph in section 4.1.1 (p11, l354-358). ‘This suggests the phosphorylation of PKA substrates to occur within a restricted nano-compartment of 15-25 nm range inside the cell [94]. Therefore, PKA subcellular compartmentalization appears essential to ensure selective phosphorylation of its substrates (see section 4.)’.

In section 4.2.2, there is a paragraph that is duplicated.  Only one of these is needed.

Author response: We thank the reviewer for point this out. We have now deleted the extra paragraph.

Reviewer 3 Report

The review “Cardiac cAMP-PKA signalling compartmentalization in myocardial infarction” highlights the serious issue of therapies for myocardial infarction.

In particular, the authors start discussing on the β-blockers as the cornerstone strategy for patients with MI. Although these drugs are effective, the cure and the mortality of the disease are still high. Also, β-blockers are contraindicated in some of patients and induced several side effects.

So, the authors focused their review on the characterization of therapeutic alternative to β-blockers.

They well describe the cAMP signalling and its compartmentalization and offer in detail a panorama of the new therapeutic options that could target the cAMP signalling. These potential innovative curatives could show similar benefits of β-blocker without induce their side effects.

The authors, also, are able to describe deeply and clearly the molecular mechanisms that are involved in the cAMP signaling compartmentalization in cardiac physiology and post-MI conditions.

All the paragraphs are well structured and exhaustive in the main contents and the figures and the tables are sufficiently clear.

The reference section is quite pertinent, even though should be more up-to-date, where possible.

Author Response

Revision of ms. no. cells-1166000

Point-by-point response to the reviewers:

We would like to thank reviewers for their comments, which we have addressed in full as specified below, and which we think has significantly improved the review.

Specific comments (please note that line# references may have shifted somewhat in the revised text):

Reviewer #3:

The review “Cardiac cAMP-PKA signalling compartmentalization in myocardial infarction” highlights the serious issue of therapies for myocardial infarction.

In particular, the authors start discussing on the β-blockers as the cornerstone strategy for patients with MI. Although these drugs are effective, the cure and the mortality of the disease are still high. Also, β-blockers are contraindicated in some of patients and induced several side effects.

So, the authors focused their review on the characterization of therapeutic alternative to β-blockers.

They well describe the cAMP signalling and its compartmentalization and offer in detail a panorama of the new therapeutic options that could target the cAMP signalling. These potential innovative curatives could show similar benefits of β-blocker without induce their side effects.

The authors, also, are able to describe deeply and clearly the molecular mechanisms that are involved in the cAMP signaling compartmentalization in cardiac physiology and post-MI conditions.

All the paragraphs are well structured and exhaustive in the main contents and the figures and the tables are sufficiently clear.

The reference section is quite pertinent, even though should be more up-to-date, where possible.

Author response: We appreciate that reviewer think we provided a well-structured and exhaustive review with clear figures. In our review we tried to cite the original studies. However, we followed Reviewer suggestion and modified some old references by more up-to-date ones. Therefore, we have approximately more than 50% cited refences for which studies have been published within the last 10 years (ref changes are indicated in red in the revised manuscript).